# Span Recovery for Deep Neural Networks with Applications to Input Obfuscation

**Rajesh Jayaram**
Computer Science Department
Carnegie Mellon University
Pittsburgh, PA 15213, USA
rkjayara@cs.cmu.edu

**David Woodruff**
Computer Science Department
Carnegie Mellon University
Pittsburgh, PA 15213, USA
dwoodruf@cs.cmu.edu

**Qiuyi Zhang**
Google Brain
qiuyiz@google.com

## Abstract

The tremendous success of deep neural networks has motivated the need to better understand the fundamental properties of these networks, but many of the theoretical results proposed have only been for shallow networks. In this paper, we study an important primitive for understanding the meaningful input space of a deep network: *span recovery*. For $k < n$, let $\mathbf{A} \in \mathbb{R}^{k \times n}$ be the innermost weight matrix of an arbitrary feed forward neural network $M : \mathbb{R}^n \to \mathbb{R}$, so $M(x)$ can be written as $M(x) = \sigma(\mathbf{A}x)$, for some network $\sigma : \mathbb{R}^k \to \mathbb{R}$. The goal is then to recover the row span of $\mathbf{A}$ given only oracle access to the value of $M(x)$. We show that if $M$ is a multi-layered network with ReLU activation functions, then partial recovery is possible: namely, we can provably recover $k/2$ linearly independent vectors in the row span of $\mathbf{A}$ using poly$(n)$ non-adaptive queries to $M(x)$. Furthermore, if $M$ has differentiable activation functions, we demonstrate that *full* span recovery is possible even when the output is first passed through a sign or $0/1$ thresholding function; in this case our algorithm is adaptive. Empirically, we confirm that full span recovery is not always possible, but only for unrealistically thin layers. For reasonably wide networks, we obtain full span recovery on both random networks and networks trained on MNIST data. Furthermore, we demonstrate the utility of span recovery as an attack by inducing neural networks to misclassify data obfuscated by controlled random noise as sensical inputs.

## 1 Introduction

Consider the general framework in which we are given an unknown function $f : \mathbb{R}^n \to \mathbb{R}$, and we want to learn properties about this function given only access to the value $f(x)$ for different inputs $x$. There are many contexts where this framework is applicable, such as blackbox optimization in which we are learning to optimize $f(x)$ (Djolonga et al., 2013), PAC learning in which we are learning to approximate $f(x)$ (Denis, 1998), adversarial attacks in which we are trying to find adversarial inputs to $f(x)$ (Szegedy et al., 2013), or structure recovery in which we are learning the structure of $f(x)$. For example in the case when $f(x)$ is a neural network, one might want to recover the underlying weights or architecture (Arora et al., 2014; Zhang et al., 2017). In this work, we consider the setting when $f(x) = M(x)$ is a neural network that admits a latent low-dimensional structure, namely $M(x) = \sigma(\mathbf{A}x)$ where $\mathbf{A} \in \mathbb{R}^{k \times n}$ is a rank $k$ matrix for some $k < n$, and $\sigma : \mathbb{R}^k \to \mathbb{R}$ is some neural network. In this setting, we focus primarily on the goal of recovering the *row-span* of the weight matrix $\mathbf{A}$. We remark that all our results generalize in a straightforward manner to the case when $\mathbf{A}$ is rank $r < k$.

Span recovery of general functions $f(x) = g(\mathbf{A}x)$, where $g$ is arbitrary, has been studied in some contexts, and is used to gain important information about the underlying function $f$. By learning Span($\mathbf{A}$), we in essence are capturing the relevant subspace of the input to $f$; namely, $f$ behaves identically on $x$ as it does on the projection of $x$ onto the row-span of $\mathbf{A}$. In statistics, this is known as *effective dimension reduction* or the *multi-index model* Li (1991); Xia et al. (2002). Another important motivation for span recovery is for designing *adversarial attacks*. Given the span of $\mathbf{A}$, we compute the kernel of $\mathbf{A}$, which can be used to fool the function into behaving incorrectly on inputs which are perturbed by vectors in the kernel. Specifically, if $x$ is a legitimate input correctly classified by $f$ and $y$ is a large random vector in the kernel of $\mathbf{A}$, then $x + y$ will be indistinguishable from noise but we will have $f(x) = f(x + y)$.

Several works have considered the problem from an *approximation-theoretic* standpoint, where the goal is to output a hypothesis function $\widetilde{f}$ which approximates $f$ well on a bounded domain. For instance, in the case that $\mathbf{A} \in \mathbb{R}^n$ is a rank 1 matrix and $g(\mathbf{A}x)$ is a smooth function with bounded derivatives, Cohen et al. (2012) gives an adaptive algorithm to approximate $f$. Their results also give an approximation $\widetilde{\mathbf{A}}$ to $\mathbf{A}$, under the assumption that $\mathbf{A}$ is a stochastic vector ($\mathbf{A}_i \geq 0$ for each $i$ and $\sum_i \mathbf{A}_i = 1$). Extending this result to more general rank $k$ matrices $\mathbf{A} \in \mathbb{R}^{k \times n}$, Tyagi & Cevher (2014) and Fornasier et al. (2012) give algorithms with polynomial sample complexity to find approximations $\widetilde{f}$ to twice differentiable functions $f$. However, their results do not provide any guarantee that the original matrix $\mathbf{A}$ itself or a good approximation to its span will be recovered. Specifically, the matrix $\widetilde{\mathbf{A}}$ used in the hypothesis function $\widetilde{f}(x) = \widetilde{g}(\widetilde{\mathbf{A}}x)$ of Tyagi & Cevher (2014) only has moderate correlation with the true row span of $\mathbf{A}$, and always admits some constant factor error (which can translate into very large error in any subspace approximation).

Furthermore, all aforementioned works require the strong assumption that the matrix of gradients is well-conditioned (and full rank) in order to obtain good approximations $\widetilde{f}$. In contrast, when $f(x)$ is a non-differentiable ReLU deep network with only mild assumptions on the weight matrices, we prove that the gradient matrix has rank at least $k/2$, which significantly strengthens span recovery guarantees since we do not make any assumptions on the gradient matrix. Finally, Hardt & Woodruff (2013) gives an adaptive approximate span recovery algorithm with poly($n$) samples under the assumption that the function $g$ satisfies a norm-preserving condition, which is restrictive and need not (and does not) hold for the deep neural networks we consider here.

On the empirical side, the experimental results of Tyagi & Cevher (2014) for function approximation were only carried out for simple one-layer functions, such as the logistic function in one dimension (where $\mathbf{A}$ has rank $k = 1$), and on linear functions $g^T(\mathbf{A}x + b)$, where $g \in \mathbb{R}^k$ has i.i.d. Gaussian entries. Moreover, their experiments only attempted to recover approximations $\widetilde{\mathbf{A}}$ to $\mathbf{A}$ when $\mathbf{A}$ was orthonormal. In addition, Fornasier et al. (2012) experimentally considers the approximation problem for $f$ when $f(\mathbf{A}x)$ is a third degree polynomial of the input. This leaves an experimental gap in understanding the performance of span recovery algorithms on non-smooth, multi-layer deep neural networks.

When $f(x)$ is a neural network, there have been many results that allow for weight or architecture recovery under additional assumptions; however nearly all such results are for shallow networks. Arora et al. (2014) shows that layer-wise learning can recover the architecture of random sparse neural networks. Janzamin et al. (2015) applies tensor methods to recover the weights of a two-layer neural network with certain types of smooth activations and vector-valued output, whereas Ge et al. (2019); Bakshi et al. (2019) obtain weight recovery for ReLU activations. Zhang et al. (2017) shows that SGD can learn the weights of two-layer neural networks with some specific activations. There is also a line of work for *improperly* two-layer networks, where the algorithm outputs an arbitrary hypothesis function which behaves similarly to the network under a fixed distribution (Goel et al., 2017; Goel & Klivans, 2019).

Learning properties of the network can also lead to so-called model extraction attacks or enhance classical adversarial attacks on neural networks (Jagielski et al., 2019). Adversarial attacks are often differentiated into two settings, the *whitebox* setting where the trained network weights are known or the *blackbox* setting where the network weights are unknown but attacks can still be achieved on external information, such as knowledge of the dataset, training algorithm, network architecture, or network predictions. Whitebox attacks are well-studied and usually use explicit gradients or optimization procedures to compute adversarial inputs for various tasks such as classification and reinforcement learning (Szegedy et al., 2013; Huang et al., 2017; Goodfellow et al., 2014). However, blackbox attacks are more realistic and it is clear that model recovery can enhance these attacks. The work of Papernot et al. (2017) attacks practical neural networks upon observations of predictions of adaptively chosen inputs, trains a substitute neural network on the observed data, and applies a whitebox attack on the substitute. This setting, nicknamed the practical blackbox setting (Chen et al., 2017), is what we work in, as we only observe adaptively chosen predictions without knowledge of the network architecture, dataset, or algorithms. We note that perhaps surprisingly, some of our algorithms are in fact entirely non-adaptive.

## 1.1 OUR CONTRIBUTIONS

In this paper, we provably show that span recovery for deep neural networks with high precision can be efficiently accomplished with poly($n$) function evaluations, even when the networks have poly($n$) layers and the output of the network is a scalar in some finite set. Specifically, for deep networks $M(x) : \mathbb{R}^n \to \mathbb{R}$ with ReLU activation functions, we prove that we can recover a subspace $V \subset \text{Span}(A)$ of dimension at least $k/2$ with polynomially many non-adaptive queries.[1] First, we use a volume bounding technique to show that a ReLU network has sufficiently large piece-wise linear sections and that gradient information can be derived from function evaluations. Next, by using a

---

[1]We switch to the notation $M(x)$ instead of $f(x)$ to illustrate that $M(x)$ is a neural network, whereas $f(x)$ was used to represent a general function with low-rank structure.

novel combinatorial analysis of the sign patterns of the ReLU network along with facts in polynomial algebra, we show that the gradient matrix has sufficient rank to allow for partial span recovery.

**Theorem 3.4 (informal)** *Suppose we have the network $M(x) = w^T \phi(\mathbf{W}_1 \phi(\mathbf{W}_2 \phi(\ldots \mathbf{W}_d \phi(\mathbf{A}x))\ldots))$, where $A \in \mathbb{R}^{k \times n}$ is rank $k$, $\phi$ is the ReLU and $\mathbf{W}_i \in \mathbb{R}^{k_i \times k_{i+1}}$ are weight matrices, with $k_i$ possibly much smaller than $k$. Then, under mild assumptions, there is a non-adaptive algorithm that makes $O(kn \log k)$ queries to $M(x)$ and returns in poly$(n, k)$-time a subspace $V \subseteq span(A)$ of dimension at least $\frac{k}{2}$ with probability $1 - \delta$.*

We remark that span recovery of the first weight layer is provably feasible even in the surprising case when the neural network has many "bottleneck" layers with small $O(\log(n))$ width. Because this does not hold in the linear case, this implies that the non-linearities introduced by activations in deep learning allow for much more information to be captured by the model. Moreover, our algorithm is non-adaptive, which means that the points $x_i$ at which $M(x_i)$ needs to be evaluated can be chosen in advance and span recovery will succeed with high probability. This has the benefit of being parallelizable, and possibly more difficult to detect when being used for an adversarial attack. In addition, we note that this result generalize to the case when $\mathbf{A}$ is rank $r < k$, in which setting our guarantee will instead be that we recover a subspace of dimension at least $\frac{r}{2}$ contained within the span of $A$.

In contrast with previous papers, we do not assume that the gradient matrix has large rank; rather our main focus and novelty is to prove this statement under minimal assumptions. We require only two mild assumptions on the weight matrices. The first assumption is on the *orthant probabilities* of the matrix $\mathbf{A}$, namely that the distribution of sign patterns of a vector $\mathbf{A}g$, where $g \sim \mathcal{N}(0, \mathbb{I}_n)$, is not too far from uniform. Two examples of matrices which satisfy this property are random matrices and matrices with nearly orthogonal rows. The second assumption is a non-degeneracy condition on the matrices $\mathbf{W}_i$, which enforces that products of rows of the matrices $\mathbf{W}_i$ result in vectors with non-zero coordinates.

Our next result is to show that *full* span recovery is possible for thresholded networks $M(x)$ with twice differentiable activation functions in the inner layers, when the network has a $0/1$ threshold function in the last layer and becomes therefore non-differentiable, i.e., $M(x) \in \{0, 1\}$. Since the activation functions can be arbitrarily non-linear, our algorithm only provides an approximation of the true subspace Span$(\mathbf{A})$, although the distance between the subspace we output and Span$(\mathbf{A})$ can be made exponentially small. We need only assume bounds on the first and second derivatives of the activation functions, as well as the fact that we can find inputs $x \in \mathbb{R}^n$ such that $M(x) \neq 0$ with good probability, and that the gradients of the network near certain points where the threshold evaluates to one are not arbitrarily small. We refer the reader to Section 4 for further details on these assumptions. Under these assumptions, we can apply a novel gradient-estimation scheme to approximately recover the gradient of $M(x)$ and the span of $\mathbf{A}$.

**Theorem 4.3 (informal)** *Suppose we have the network $M(x) = \tau(\sigma(\mathbf{A}x))$, where $\tau : \mathbb{R} \to \{0, 1\}$ is a threshold function and $\sigma : \mathbb{R}^k \to \mathbb{R}$ is a neural network with twice differentiable activation functions, and such that $M$ satisfies the conditions sketched above (formally defined in Section 4). Then there is an algorithm that runs in poly$(N)$ time, making at most poly$(N)$ queries to $M(x)$, where $N = poly(n, k, \log(\frac{1}{\epsilon}), \log(\frac{1}{\delta}))$, and returns with probability $1 - \delta$ a subspace $V \subset \mathbb{R}^n$ of dimension $k$ such that for any $x \in V$, we have*

$$\|\mathbf{P}_{Span(\mathbf{A})}x\|_2 \geq (1 - \epsilon)\|x\|_2$$

*where $\mathbf{P}_{Span(\mathbf{A})}$ is the orthogonal projection onto the span of $\mathbf{A}$.*

Empirically, we verify our theoretical findings by running our span recovery algorithms on randomly generated networks and trained networks. First, we confirm that full recovery is not possible for all architectures when the network layer sizes are small. This implies that the standard assumption that the gradient matrix is full rank does not always hold. However, we see that realistic network architectures lend themselves easily to full span recovery on both random and trained instances. We emphasize that this holds even when the network has many small layers, for example a ReLU network that has 6 hidden layers with $[784, 80, 40, 30, 20, 10]$ nodes, in that order, can still admit full span recovery of the rank 80 weight matrix.

Furthermore, we observe that we can effortlessly apply input obfuscation attacks after a successful span recovery and cause misclassifications by tricking the network into classifying noise as normal inputs with high confidence. Specifically, we can inject large amounts of noise in the null space of $\mathbf{A}$ to arbitrarily obfuscate the input without changing the output of the network. We demonstrate the utility of this attack on MNIST data, where we use span recovery to generate noisy images that are classified by the network as normal digits with high confidence. We note that this veers away from traditional adversarial attacks, which aim to drastically change the network output with humanly-undetectable changes in the input. In our case, we attempt the arguably more challenging problem of drastically changing the input without affecting the output of the network.

## 2 PRELIMINARIES

**Notation** For a vector $x \in \mathbb{R}^k$, the *sign pattern* of $x$, denoted $\text{sign}(x) \in \{0,1\}^k$, is the indicator vector for the non-zero coordinates of $x$. Namely, $\text{sign}(x)_i = 1$ if $x_i \neq 0$ and $\text{sign}(x)_i = 0$ otherwise. Given a matrix $\mathbf{A} \in \mathbb{R}^{n \times m}$, we denote its singular values as $\sigma_{\min}(\mathbf{A}) = \sigma_{\min\{n,m\}}, \ldots, \sigma_1(\mathbf{A}) = \sigma_{\max}(\mathbf{A})$. The *condition number* of $\mathbf{A}$ is denoted $\kappa(\mathbf{A}) = \sigma_{\max}(\mathbf{A})/\sigma_{\min}(\mathbf{A})$. We let $\mathbb{I}_n \in \mathbb{R}^{n \times n}$ denote the $n \times n$ identity matrix. For a subspace $V \subset \mathbb{R}^n$, we write $\mathbf{P}_V \in \mathbb{R}^{n \times n}$ to denote the orthogonal projection matrix onto $V$. If $\mu \in \mathbb{R}^n$ and $\Sigma \in \mathbb{R}^{n \times n}$ is a PSD matrix, we write $\mathcal{N}(\mu, \Sigma)$ to denote the multi-variate Gaussian distribution with mean $\mu$ and covariance $\Sigma$.

**Gradient Information** For any function $f(x) = g(\mathbf{A}x)$, note that $\nabla f(x) = \mathbf{A}^\top g(\mathbf{A}x)$ must be a vector in the row span of $\mathbf{A}$. Therefore, span recovery boils down to understanding the span of the gradient matrix as $x$ varies. Specifically, note that if we can find points $x_1, .., x_k$ such that $\{\nabla f(x_i)\}$ are linearly independent, then the full span of $\mathbf{A}$ can be recovered using the span of the gradients. To our knowledge, previous span recovery algorithms heavily rely on the assumption that the gradient matrix is full rank and in fact well-conditioned. Specifically, for some distribution $\mathcal{D}$, it is assumed that $H_f = \int_{x \sim \mathcal{D}} \nabla f(x) \nabla f(x)^\top \, dx$ is a rank $k$ matrix with a minimum non-zero singular value bounded below by $\alpha$ and the number of gradient or function evaluations needed depends inverse polynomially in $\alpha$. In contrast, in this paper, when $f(x)$ is a neural network, we provably show that $H_f$ is a matrix of sufficiently high rank or large minimum non-zero singular value under mild assumptions, using tools in polynomial algebra.

## 3 DEEP NETWORKS WITH RELU ACTIVATIONS

In this section, we demonstrate that partial span recovery is possible for deep ReLU networks. Specifically, we consider neural networks $M(x) : \mathbb{R}^n \to \mathbb{R}$ of the form

$$M(x) = w^T \phi(\mathbf{W}_1 \phi(\mathbf{W}_2 \phi(\ldots \mathbf{W}_d \phi(\mathbf{A}x)) \ldots)$$

where $\phi(x)_i = \max\{x_i, 0\}$ is the RELU (applied coordinate-wise to each of its inputs), and $\mathbf{W}_i \in \mathbb{R}^{k_i \times k_{i+1}}$, and $w \in \mathbb{R}^{k_d}$, and $\mathbf{A}$ has rank $k$. We note that $k_i$ can be much smaller than $k$. In order to obtain partial span recovery, we make the following assumptions parameterized by a value $\gamma > 0$ (our algorithms will by polynomial in $1/\gamma$):

- **Assumption 1:** For every sign pattern $S \in \{0,1\}^k$, we have $\mathbf{Pr}_{g \sim \mathcal{N}(0, I_n)} \left[ \text{sign}(\phi(\mathbf{A}g)) = S \right] \geq \gamma/2^k$.
- **Assumption 2:** For any $S_1, \ldots, S_d \neq \emptyset$ where $S_i \subseteq [k_i]$, we have $w^T \left( \prod_{i=1}^d (\mathbf{W}_i)_{S_i} \right) \in \mathbb{R}^k$ is entry-wise non-zero. Here $(\mathbf{W}_i)_{S_i}$ is the matrix with the rows $j \notin S_i$ set equal to 0. Moreover, we assume
$$\mathbf{Pr}_{g \sim \mathcal{N}(0, I_k)} \left[ M(g) = 0 \right] \leq \frac{\gamma}{8}.$$

Our first assumption is an assumption on the *orthant probabilities* of the distribution $\mathbf{A}g$. Specifically, observe that $\mathbf{A}g \in \mathbb{R}^k$ follows a multi-variate Gaussian distribution with covariance matrix $\mathbf{A}\mathbf{A}^T$. Assumption 1 then states that the probability that a random vector $x \sim \mathcal{N}(0, \mathbf{A}\mathbf{A}^T)$ lies in a certain orthant of $\mathbb{R}^k$ is not too far from uniform. We remark that orthant probabilities of multivariate Gaussian distributions are well-studied (see e.g., Miwa et al. (2003); Bacon (1963); Abrahamson et al. (1964)), and thus may allow for the application of this assumption to a larger class of matrices. In particular, we show it is satisfied by both random matrices and orthogonal matrices. Our second assumption is a non-degeneracy condition on the weight matrices $\mathbf{W}_i$ – namely, that products of $w^T$ with non-empty sets of rows of the $\mathbf{W}_i$ result in entry-wise non-zero vectors. In addition, Assumption 2 requires that the network is non-zero with probability that is not arbitrarily small, otherwise we cannot hope to find even a single $x$ with $M(x) \neq 0$.

In the following lemma, we demonstrate that these conditions are satisfied by randomly initialized networks, even when the entries of the $\mathbf{W}_i$ are not identically distributed.

**Lemma 3.1.** *If $\mathbf{A} \in \mathbb{R}^{k \times n}$ is an arbitrary matrix with orthogonal rows, or if $n > \Omega(k^3)$ and $\mathbf{A}$ has entries that are drawn i.i.d. from some sub-Gaussian distribution $\mathcal{D}$ with expectation $0$, unit variance, and constant sub-Gaussian norm $\|\mathcal{D}\|_{\psi_2} = \sup_{p \geq 1} p^{-1/2} \left( \mathbf{E}_{X \sim \mathcal{D}} |X|^p \right)^{1/p}$ then with probability at least $1 - e^{-k^2}$, $\mathbf{A}$ satisfies Assumption 1 with $\gamma \geq 1/2$. Moreover, if the weight matrices $w, \mathbf{W}_1, \mathbf{W}_2, ..., \mathbf{W}_d$ with $\mathbf{W}_i \in \mathbb{R}^{k_i \times k_{i+1}}$ have entries that are drawn independently (and possibly non-identically) from continuous symmetric distributions, and if $k_i \geq \log(\frac{16d}{\delta\gamma})$ for each $i \in [d]$, then Assumption 2 holds with probability $1 - \delta$.*

### 3.1 ALGORITHM FOR SPAN RECOVERY

The algorithm for recovery is given in Algorithm 1. Our algorithm computes the gradient $\nabla M(g_i)$ for different Gaussian vectors $g_i \sim \mathcal{N}(0, I_k)$, and returns the subspace spanned by these gradients. To implement this procedure,

---

**Algorithm 1:** Span Recovery with Non-Adaptive Gradients

---

**Input:** function $M(x) : \mathbb{R}^n \to \mathbb{R}$, $k$: latent dimension , $\gamma$: probability parameter

**1** Set $r = O(k \log(k)/\gamma)$

**2 for** $i = 0, \dots, r$ **do**

**3**    Generate random Gaussian vector $g^i \sim \mathcal{N}(0, I_n)$.

**4**    **Compute Gradient:** $z^i = \nabla M(g^i)$                                     ▷ Lemma 3.2

**5 end**

**6 return** $Span(z^1, z^2, \dots, z^r)$

---

we must show that it is possible to compute gradients via the perturbational method (i.e. finite differences), given only oracle queries to the network $M$. Namely, we firstly must show that if $g \sim \mathcal{N}(0, \mathbb{I}_n)$ then $\nabla M(g)$ exists, and moreover, that $\nabla M(x)$ exists for all $x \in \mathcal{B}_\epsilon(g)$, where $\mathcal{B}_\epsilon(g)$ is a ball of radius $\epsilon$ centered at $g$, and $\epsilon$ is some value with polynomial bit complexity which we can bound. To demonstrate this, we show that for any fixing of the sign patterns of the network, we can write the region of $\mathbb{R}^n$ which satisfies this sign pattern and is $\epsilon$-close to one of the $O(dk)$ ReLU thresholds of the network as a linear program. We then show that the feasible polytope of this linear program is contained inside a Euclidean box in $\mathbb{R}^n$, which has one side of length $\epsilon$. Using this containment, we upper bound the volume of the polytope in $\mathbb{R}^n$ which is $\epsilon$ close to each ReLU, and union bound over all sign patterns and ReLUs to show that the probability that a Gaussian lands in one of these polytopes is exponentially small.

**Lemma 3.2.** *There is an algorithm which, given $g \sim \mathcal{N}(0, I_k)$, with probability $1 - \exp(-n^c)$ for any constant $c > 1$ (over the randomness in $g$), computes $\nabla M(g) \in \mathbb{R}^n$ with $O(n)$ queries to the network, and in $poly(n)$ runtime.*

Now observe that the gradients of the network lie in the row-span of $\mathbf{A}$. To see this, for a given input $x \in \mathbb{R}^n$, let $S_0(x) \in \mathbb{R}^k$ be the sign pattern of $\phi(\mathbf{A}x) \in \mathbb{R}^k$, and more generally define $S_i(x) \in \mathbb{R}^{k_i}$ via

$$S_i(x) = \text{sign}\Big(\phi(\mathbf{W}_i \phi(\mathbf{W}_{i+1} \phi(\dots \mathbf{W}_d \phi(Ax)) \dots))\Big)$$

Then $\nabla M(x) = (w^T \cdot (\prod_{i=1}^d (\mathbf{W}_i)_{S_i}))) \mathbf{A}_{S_0}$, which demonstrates the claim that the gradients lie in the row-span of $\mathbf{A}$. Now define $z^i = \nabla M(g^i)$ where $g^i \sim \mathcal{N}(0, \mathbb{I}_n)$, and let $\mathbf{Z}$ be the matrix where the $i$-th row is equal to $z_i$. We will prove that $\mathbf{Z}$ has rank at least $k/2$. To see this, first note that we can write $\mathbf{Z} = \mathbf{VA}$, where $\mathbf{V}$ is some matrix such that the non-zero entries in the $i$-th row are precisely the coordinates in the set $S_0^i$, where $S_j^i = S_j(g^i)$ for any $j = 0, 1, 2, \dots, d$ and $i = 1, 2, \dots, r$. We first show that $\mathbf{V}$ has rank at least $ck$ for a constant $c > 0$. To see this, suppose we have computed $r$ gradients so far, and the rank of $\mathbf{V}$ is less than $ck$ for some $0 < c < 1/2$. Now $\mathbf{V} \in \mathbb{R}^{r \times k}$ is a fixed rank-$ck$ matrix, so the span of the matrix can be expressed as a linear combination of some fixed subset of $ck$ of its rows. We use this fact to show in the following lemma that the set of all possible sign patterns obtainable in the row span of $\mathbf{V}$ is much smaller than $2^k$. Thus a gradient $z^{r+1}$ with a uniform (or nearly uniform) sign pattern will land outside this set with good probability, and thus will increase the rank of $\mathbf{Z}$ when appended.

**Lemma 3.3.** *Let $\mathbf{V} \in \mathbb{R}^{r \times k}$ be a fixed at most rank $ck$ matrix for $c \leq 1/2$. Then the number of sign patterns $S \subset [k]$ with at most $k/2$ non-zeros spanned by the rows of $\mathbf{V}$ is at most $\frac{2^k}{\sqrt{k}}$. In other words, the set $\mathcal{S}(\mathbf{V}) = \{sign(w) \mid w \in span(\mathbf{V}), nnz(w) \leq \frac{k}{2}\}$ has size at most $\frac{2^k}{\sqrt{k}}$.*

**Theorem 3.4.** *Suppose the network $M(x) = w^T \phi(W_1 \phi(W_2 \phi(\dots W_d \phi(Ax)) \dots))$, where $\phi$ is the ReLU, satisfies **Assumptions 1 and 2**. Then the algorithm given in Figure 1 makes $O(kn \log(k/\delta)/\gamma)$ queries to $M(x)$ and returns in $poly(n, k, 1/\gamma, \log(1/\delta))$ time a subspace $V \subseteq span(A)$ of dimension at least $\frac{k}{2}$ with probability $1 - \delta$.*

## 4    NETWORKS WITH THRESHOLDING ON DIFFERENTIABLE ACTIVATIONS

In this section, we consider networks that have a threshold function at the output node, as is done often for classification. Specifically, let $\tau : \mathbb{R} \to \{0, 1\}$ be the threshold function: $\tau(x) = 1$ if $x \geq 1$, and $\tau(x) = 0$ otherwise. Again, we let $\mathbf{A} \in \mathbb{R}^{k \times n}$ where $k < n$, be the innermost weight matrix. The networks $M : \mathbb{R}^n \to \mathbb{R}$ we consider are then of the form:

$$M(x) = \tau(\mathbf{W}_1 \phi_1(\mathbf{W}_2 \phi_2(\dots \phi_d \mathbf{A}x)) \dots)$$

where $\mathbf{W}_i \in \mathbb{R}^{k_i \times k_{i+1}}$ and each $\phi_i$ is a continuous, differentiable activation function applied entrywise to its input. We will demonstrate that even for such functions with a binary threshold placed at the end, giving us minimal information about the network, we can still achieve *full* span recovery of the weight matrix $\mathbf{A}$, albeit with the cost of an $\epsilon$-approximation to the subspace. Note that the latter fact is inherent, since the gradient of any function that is not linear

in some ball around each point cannot be obtained exactly without infinitely small perturbations of the input, which we do not allow in our model.

We can simplify the above notation, and write $\sigma(x) = \mathbf{W}_1\phi_1(\mathbf{W}_2\phi_2(\ldots\phi_d\mathbf{A}x))\ldots)$, and thus $M(x) = \tau(\sigma(x))$. Our algorithm will involve building a subspace $V \subset \mathbb{R}^n$ which is a good approximation to the span of $\mathbf{A}$. At each step, we attempt to recover a new vector which is very close to a vector in $\mathrm{Span}(\mathbf{A})$, but which is nearly orthogonal to the vectors in $V$. Specifically, after building $V$, on an input $x \in \mathbb{R}^n$, we will query $M$ for inputs $M((\mathbb{I}_n - \mathbf{P}_V)x)$. Recall that $\mathbf{P}_V$ is the projection matrix onto $V$, and $\mathbf{P}_{V\perp}$ is the projection matrix onto the subspace orthogonal to $V$. Thus, it will help here to think of the functions $M, \sigma$ as being functions of $x$ and not $(\mathbb{I}_n - \mathbf{P}_V)x$, and so we define $\sigma_V(x) = \sigma(\mathbf{A}(\mathbb{I}_n - \mathbf{P}_V)x)$, and similarly $M_V(x) = \tau(\sigma_V(x))$. For the results of this section, we make the following assumptions on the activation functions.

**Assumptions:**

1. The function $\phi_i : \mathbb{R} \to \mathbb{R}$ is continuous and twice differentiable, and $\phi_i(0) = 0$.
2. $\phi_i$ and $\phi_i'$ are $L_i$-Lipschitz, meaning:

$$\sup_{x\in\mathbb{R}}\left|\frac{d}{dx}\phi_i(x)\right| \leq L_i, \quad \sup_{x\in\mathbb{R}}\left|\frac{d^2}{d^2x}\phi_i(x)\right| \leq L_i$$

3. The network is non-zero with bounded probability: for every subspace $V \subset \mathbb{R}^n$ of dimension $\dim(V) < k$, we have that $\mathbf{Pr}_{g\sim\mathcal{N}(0,\mathbb{I}_n)}[\sigma_V(g) \geq 1] \geq \gamma$ for some value $\gamma > 0$.
4. Gradients are not arbitrarily small near the boundary: for every subspace $V \subset \mathbb{R}^n$ of dimension $\dim(V) < k$

$$\mathbf{Pr}_{g\sim\mathcal{N}(0,\mathbb{I}_n)}[\,|\nabla_g\sigma_V(cg)| \geq \eta, \forall c > 0 \text{ such that } \sigma_V(cg) = 1, \text{ and } \sigma_V(g) \geq 1] \geq \gamma$$

for some values $\eta, \gamma > 0$, where $\nabla_g\sigma_V(cg)$ is the directional derivative of $\sigma_V$ in the direction $g$.

The first two conditions are standard and straightforward, namely $\phi_i$ is differentiable, and has bounded first and second derivatives (note that for our purposes, they need only be bounded in a ball of radius $\mathrm{poly}(n)$). Since $M(x)$ is a threshold applied to $\sigma(x)$, the third condition states that it is possible to find inputs $x$ with non-zero network evaluation $M(x)$. Our condition is slightly stronger, in that we would like this to be possible even when $x$ is projected away from any $k' < k$ dimensional subspace (note that this ensures that $\mathbf{A}x$ is non-zero, since $\mathbf{A}$ has rank $k$).

The last condition simply states that if we pick a random direction $g$ where the network is non-zero, then the gradients of the network are not arbitrarily small along that direction at the threshold points where $\sigma(c \cdot g) = 1$. Observe that if the gradients at such points are vanishingly small, then we cannot hope to recover them. Moreover, since $M$ only changes value at these points, these points are the only points where information about $\sigma$ can be learned. Thus, the gradients at these points are the *only* gradients which could possibly be learned. We note that the running time of our algorithms will be polynomial in $\log(1/\eta)$, and thus we can even allow the gradient size $\eta$ to be exponentially small.

## 4.1 THE APPROXIMATE SPAN RECOVERY ALGORITHM

We now formally describe and analyze our span recovery algorithm for networks with differentiable activation functions and $0/1$ thresholding. Let $\kappa_i$ be the condition number of the $i$-th weight matrix $\mathbf{W}_i$, and let $\delta > 0$ be a failure probability, and let $\epsilon > 0$ be a precision parameter which will affect the how well the subspace we output will approximate $\mathrm{Span}(\mathbf{A})$. Now fix $N = \mathrm{poly}(n, k, \frac{1}{\gamma}, \sum_{i=1}^d \log(L_i), \sum_{i=1}^d \log(\kappa_i), \log(\frac{1}{\eta}), \log(\frac{1}{\epsilon}), \log(\frac{1}{\delta}))$. The running time and query complexity of our algorithm will be polynomial in $N$. Our algorithm for approximate span recovery is given formally in Algorithm 2.

**Proposition 4.1.** *Let $V \subset \mathbb{R}^n$ be a subspace of dimension $k' < k$, and fix any $\epsilon_0 > 0$. Then we can find a vector $x$ with $0 \leq \sigma_V(x) - 1 \leq 2\epsilon_0$ in expected $O(1/\gamma + N\log(1/\epsilon_0))$ time. Moreover, with probability $\gamma/2$ we have that $\nabla_x\sigma_V(x) > \eta/4$ and the tighter bound of $\epsilon_0\eta2^{-N} \leq \sigma_V(x) - 1 \leq 2\epsilon_0$.*

We will apply the above proposition as input to the following Lemma 4.2, which is the main technical result of this section. Our approach involves first taking the point $x$ from Proposition 4.1 such that $\sigma_V(x)$ is close but bounded away from the boundary, and generating $n$ perturbations at this point $M_V(x + u_i)$ for carefully chosen $u_i$. While we do not know the value of $\sigma_V(x + u_i)$, we can tell for a given scaling $c > 0$ if $\sigma_V(x + cu_i)$ has crossed the boundary, since we will then have $M_V(x + cu_i) = 0$. Thus, we can estimate the directional derivative $\nabla_{u_i}\sigma(x)$ by finding a value $c_i$ via a binary search such that $\sigma_V(x + c_iu_i)$ is exponentially closer to the boundary than $\sigma_V(x)$. In order for our estimate to be accurate, we must carefully upper and lower bound the gradients and Hessian of $\sigma_v$ near $x$, and demonstrate that the linear approximation of $\sigma_v$ at $x$ is still accurate at the point $x + c_iu_i$ where the boundary is crossed. Since each

---

**Algorithm 2:** Span Recovery With Adaptive Gradients

---

1   $V \leftarrow \emptyset$, $N = \text{poly}\big(n, k, \frac{1}{\gamma}, \sum_{i=1}^{d} \log(L_i), \sum_{i=1}^{d} \log(\kappa_i), \log(\frac{1}{\eta}), \log(\frac{1}{\epsilon}), \log(\frac{1}{\delta})\big)$, $\epsilon_0 \leftarrow 2^{-\text{poly}(N)}$

2   **for** $i = 0, \ldots, k$ **do**

3      Generate $g \sim \mathcal{N}(0, \mathbb{I}_n)$ until $M((\mathbb{I}_n - \mathbf{P}_V)g) = 1$

4      Find a scaling $\alpha > 0$ via binary search on values $\tau(\sigma_V(\alpha g))$ such that $x = \alpha g$ satisfies

       $\epsilon_0 \eta 2^{-N} \leq \sigma_V(x) - 1 \leq 2\epsilon_0.$                                $\triangleright$ Proposition 4.1

5      Generate $g_1, \ldots, g_n \sim \mathcal{N}(0, \mathbb{I}_n)$, and set $u_i = \big(g_i 2^{-N} - x/\|x\|_2\big).$

6      For each $i \in [n]$, binary search over values $c$ to find $c_i$ such that

$$1 - \beta \leq \sigma_V(x + c_i u_i) \leq 1$$

     where $\beta = 2^{-N^2} \epsilon_0^2.$

7      If any $c_i$ satisfies $|c_i| \geq (10 \cdot 2^{-N} \epsilon_0/\eta)$, restart from line 5 (regenerate the Gaussian $g$).

8      Otherwise, define $\mathbf{B} \in \mathbb{R}^{n \times n}$ via $\mathbf{B}_{*,i} = u_i$, where $\mathbf{B}_{*,i}$ is the $i$-th column of $\mathbf{B}$. Define $b \in \mathbb{R}^n$ by $b_i = 1/c_i.$

9      Let $y^*$ be the solution to:

$$\min_{y \in \mathbb{R}^n} \|y^T \mathbf{B} - b^T\|_2^2$$

     Set $v_i = y^*$ and $V \leftarrow \text{Span}(V, v_i).$

10   **end**

11   **return** $V$

---

value of $1/c_i$ is precisely proportional to $\nabla_{u_i} \sigma(x) = \langle \nabla \sigma(x), u_i \rangle$, we can then set up a linear system to approximately solve for the gradient $\nabla \sigma(x)$ (lines 8 and 9 of Algorithm 2).

**Lemma 4.2.** *Fix any $\epsilon, \delta > 0$, and let $N$ be defined as above. Then given any subspace $V \subset \mathbb{R}^n$ with dimension $dim(V) < k$, and given $x \in \mathbb{R}^n$, such that $\epsilon_0 \eta 2^{-N} \leq \sigma_V(x) - 1 \leq 2\epsilon_0$ where $\epsilon_0 = \Theta(2^{-N^C}/\epsilon)$ for a sufficiently large constant $C = O(1)$, and such that $\nabla_x \sigma_V(x) > \eta/2$, then with probability $1 - 2^{-N/n^2}$, we can find a vector $v \in \mathbb{R}^n$ in expected poly($N$) time, such that $\|\mathbf{P}_{Span(\mathbf{A})} v\|_2 \geq (1 - \epsilon)\|v\|_2$, and such that $\|\mathbf{P}_V v\|_2 \leq \epsilon \|v\|_2$.*

**Theorem 4.3.** *Suppose the network $M(x) = \tau(\sigma(\mathbf{A}x))$ satisfies the conditions described at the beginning of this section. Then Algorithm 2 runs in poly($N$) time, making at most poly($N$) queries to $M(x)$, where $N = \text{poly}(n, k, \frac{1}{\gamma}, \sum_{i=1}^{d} \log(L_i), \sum_{i=1}^{d} \log(\kappa_i), \log(\frac{1}{\eta}), \log(\frac{1}{\epsilon}), \log(\frac{1}{\delta}))$, and returns with probability $1 - \delta$ a subspace $V \subset \mathbb{R}^n$ of dimension $k$ such that for any $x \in V$, we have $\|\mathbf{P}_{Span(\mathbf{A})} x\|_2 \geq (1 - \epsilon)\|x\|_2$.*

## 5   EXPERIMENTS

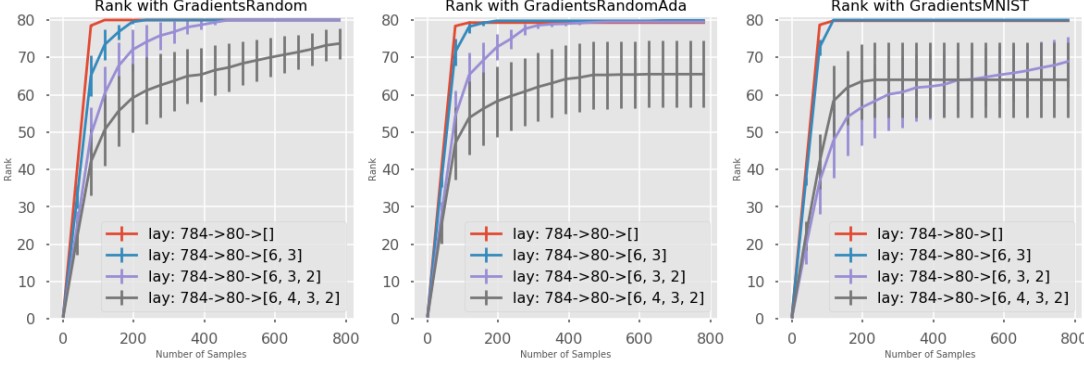

Figure 1: Partial span recovery of small networks with layer sizes specified in the legend. Note that 784->80->[6,3] indicates a 4 layer neural network with hidden layer sizes 784, 80, 6, and 3, in that order. Full span recovery is not always possible and recovery deteriorates as width decreases and depth increases.

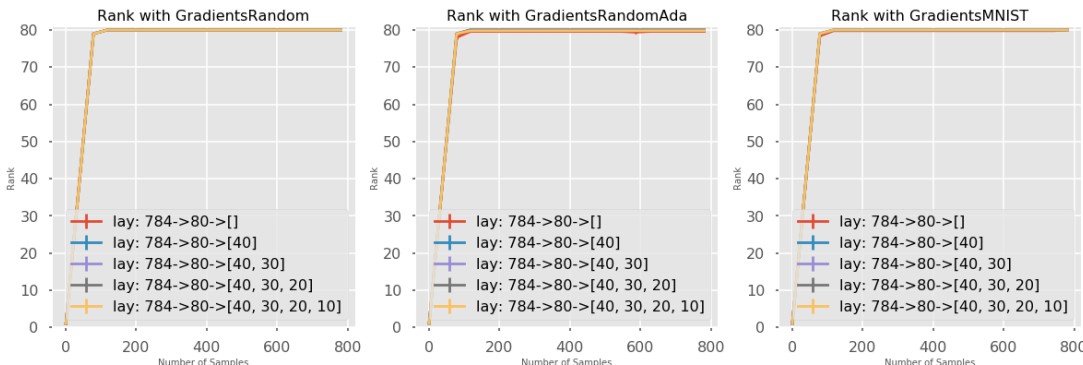

Figure 2: Full span recovery of realistic networks with moderate widths and reasonable architectures. Full recovery occurs with only 100 samples for a rank 80 weight matrix in all settings.

When applying span recovery for a given network, we first calculate the gradients analytically via auto-differentiation at a fixed number of sample points distributed according to a standard Gaussian. Our networks are feedforward, fully-connected with ReLU units; therefore, as mentioned above, using analytic gradients is as precise as using finite differences due to piecewise linearity. Then, we compute the rank of the resulting gradient matrix, where the rank is defined to be the number of singular values that are above 1e-5 of the maximum singular value. In our experiments, we attempt to recover the full span of a 784-by-80 matrix with decreasing layer sizes for varying sample complexity, as specified in the figures. For the MNIST dataset, we use a size 10 vector output and train according to the softmax cross entropy loss, but we only calculate the gradient with respect to the first output node.

Our recovery algorithms are GradientsRandom (Algorithm 1), GradientsRandomAda (Algorithm 2), and GradientsM-NIST. GradientsRandom is a direct application of our first span recovery algorithm and calculates gradients via perturbations at random points for a random network. GradientsRandomAda uses our adaptive span recovery algorithm for a random network. Finally, GradientsMNIST is an application of GradientsRandom on a network with weights trained on MNIST data. In general, we note that the experimental outcomes are very similar among all three scenarios.

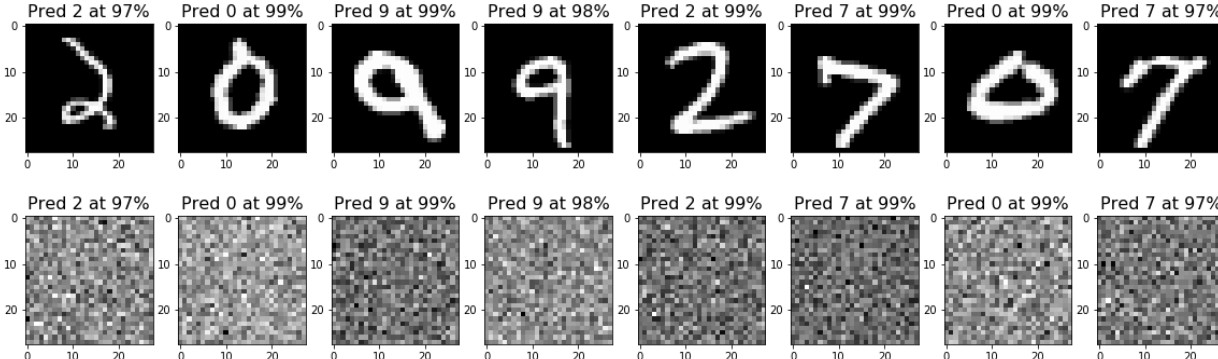

Figure 3: Fooling ReLU networks into misclassifying noise as digits by introducing Gaussian noise into the null space after span recovery. The prediction of the network is presented above the images, along with its softmax probability.

For networks with very small widths and multiple layers, we see that span recovery deteriorates as depth increases, supporting our theory (see Figure 1). This holds both in the case when the networks are randomly initialized with Gaussian weights or trained on a real dataset (MNIST) and whether we use adaptive or non-adaptive recovery algorithms. However, we note that these small networks have unrealistically small widths (less than 10) and when trained on MNIST, these networks fail to achieve high accuracy, all falling below 80 percent. The small width case is therefore only used to support, with empirical evidence, why our theory cannot possibly guarantee full span recovery under every network architecture.

For more realistic networks with moderate or high widths, however, full span recovery seems easy and implies a real possibility for attack (see Figure 2). Although we tried a variety of widths and depths, the results are robust to reasonable settings of layer sizes and depths. Therefore, we only present experimental results with sub-networks of a

network with layer sizes [784, 80, 40, 30, 20, 10]. Note that full span recovery of the first-layer weight matrix with rank 80 is achieved almost immediately in all cases, with less than 100 samples.

On the real dataset MNIST, we demonstrate the utility of span recovery algorithms as an attack to fool neural networks to misclassify noisy inputs (see Figure 3). We train a ReLU network (to around 95 percent accuracy) and recover its span by computing the span of the resulting gradient matrix. Then, we recover the null space of the matrix and generate random Gaussian noise projected onto the null space. We see that our attack successfully converts images into noisy versions without changing the output of the network, implying that allowing a full (or even partial) span recovery on a classification network can lead to various adversarial attacks despite not knowing the exact weights of the network.

ACKNOWLEDGMENTS

The authors Rajesh Jayaram and David Woodruff would like to thank the partial support by the National Science Foundation under Grant No. CCF-1815840.

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

## A  Missing Proofs from Section 3

We first restate the results which have had their proofs omitted, and include their proofs subsequently.

**Lemma 3.1**  *If $\mathbf{A} \in \mathbb{R}^{k \times n}$ has orthogonal rows, or if $n > \Omega(k^3)$ and $\mathbf{A}$ has entries that are drawn i.i.d. from some sub-Gaussian distribution $\mathcal{D}$ with expectation $0$, unit variance, and constant sub-gaussian norm $\|\mathcal{D}\|_{\psi_2} = \sup_{p \geq 1} p^{-1/2} \left(\mathbf{E}_{X \sim \mathcal{D}} |X|^p\right)^{1/p}$ then with probability at least $1 - e^{-k^2}$, $\mathbf{A}$ satisfies Assumption 1 with $\gamma \geq 1/2$.*

*Moreover, if the weight matrices $w, \mathbf{W}_1, \mathbf{W}_2, ..., \mathbf{W}_d$ with $\mathbf{W}_i \in \mathbb{R}^{k_i \times k_{i+1}}$ have entries that are drawn independently (and possibly non-identically) from continuous symmetric distributions, and if $k_i \geq \log(\frac{16d}{\delta \gamma})$ for each $i \in [d]$, then Assumption 2 holds with probability $1 - \delta$.*

*Proof.* By Theorem 5.58 of Vershynin (2010), if the entries $\mathbf{A}$ are drawn i.i.d. from some sub-Gaussian isotropic distribution $\mathcal{D}$ over $R^n$ such that $\|\mathbf{A}_j\|_2 = \sqrt{n}$ almost surely, then $\sqrt{n} - C\sqrt{k} - t \leq \sigma_{\min}(\mathbf{A}) \leq \sigma_{\max}(\mathbf{A}) \leq \sqrt{n} + C\sqrt{k} + t$ with probability at least $1 - 2e^{-ct^2}$, for some constants $c, C > 0$ depending only on $\|\mathcal{D}\|_{\psi_2}$. Since the entries are i.i.d. with variance 1, it follows that the rows of $\mathbf{A}$ are isotropic. Moreover, we can always condition on the rows having norm exactly $\sqrt{n}$, and pulling out a positive diagonal scaling through the first Relu of $M(x)$, and absorbing this scaling into $\mathbf{W}_d$. It follows that the conditions of the theorem hold, and we have $\sqrt{n} - C\sqrt{k} \leq \sigma_{\min}(\mathbf{A}) \leq \sigma_{\max}(\mathbf{A}) \leq \sqrt{n} + C\sqrt{k}$ with probability at least $1 - e^{-k^2}$ for a suitably large re scaling of the constant $C$. Setting $n > \Omega(k^3)$, it follows that $\kappa(A) < (1 + 1/(100k))$, which holds immediately if $\mathbf{A}$ has orthogonal rows.

Now observe that $\mathbf{A}g$ is distributed as a multi-variate Gaussian with co-variance $\mathbf{A}^T \mathbf{A}$, and is therefore given by the probability density function (pdf)

$$p'(x) = \frac{1}{(2\pi)^{k/2} \det(\mathbf{A}^T \mathbf{A})^{1/2}} \exp\left(-\frac{1}{2} x^T \mathbf{A}^T \mathbf{A} x\right)$$

Let $p(x) = \frac{1}{(2\pi)^{k/2}} \exp\left(-\frac{1}{2} x^T x\right)$ be the pdf of an identity covariance Gaussian $\mathcal{N}(0, I_k)$. We lower bound $p'(x)/p(x)$ for $x$ with $\|x\|_2^2 \leq 16k$. In this case, we have

$$
\begin{aligned}
\frac{p'(x)}{p(x)} &= \frac{1}{\det(\mathbf{A}^T \mathbf{A})^{1/2}} \exp\left(-\frac{1}{2}\left(x^T \mathbf{A}^T \mathbf{A} x - \|x\|_2^2\right)\right) \\
&\geq \kappa(\mathbf{A})^{-k} \exp\left(-\frac{1}{2}\left(\|x\|_2^2(1 + 1/(100k)) - \|x\|_2^2\right)\right) \\
&\geq \kappa(\mathbf{A})^{-k} \exp\left(-\frac{1}{2}\|x\|_2^2/(100k)\right) \\
&\geq (1 + 1/(100k))^{-k} \exp\left(-\frac{1}{2}(16/100)\right) \\
&\geq (1 + 1/(100k))^{-k} \exp\left(-\frac{1}{2}(16/100)\right) \\
&\geq 1/2
\end{aligned}
\tag{1}
$$

Thus for any sign pattern $S$, $\mathbf{Pr}[\text{sign}(Ag) = S \ : \ \|Ag\|_2^2 \leq k] \geq \frac{1}{2}\mathbf{Pr}[\text{sign}(g) = S \ : \ \|g\|_2^2 \leq k]$. Now $\mathbf{Pr}[\text{sign}(g) = S] = 2^{-k}$, and spherical symmetry of Gaussians, $\mathbf{Pr}[\text{sign}(g) = S \ : \ \|g\|_2^2 \leq k] = 2^{-k}$, and thus $\mathbf{Pr}[\text{sign}(Ag) = S \ : \ \|Ag\|_2^2 \leq k] \geq 2^{-k-1}$. Now $\|Ag\|_2^2 \leq 2\|g\|_2^2$ which is distributed as a $\chi^2$ random variable with $k$-degrees of freedom. By standard concentration results for $\chi^2$ distributions (Lemma 1 Laurent & Massart (2000)), we have $\mathbf{Pr}[\|g\|_2^2 \geq 8k] \leq e^{-2k}$, so $\mathbf{Pr}[\|Ag\|_2^2 \geq 16k] \leq \mathbf{Pr}[\|g\|_2^2 \geq 8k] \leq e^{-2k}$. By a union bound, $\mathbf{Pr}[\text{sign}(Ag) = S \ : \ \|Ag\|_2^2 \leq k] \geq 2^{-k-1} - e^{-2k} \geq \frac{1}{4}2^{-k}$, which completes the proof of the first claim with $\gamma = 1/4$.

For the second claim, by an inductive argument, the entries in the rows $i \in S_j$ of the product $\mathbf{W}_{S_j}\left(\prod_{i=j+1}^{d}(\mathbf{W}_i)_{S_i}\right)$ are drawn from a continuous distribution. Thus each column of $\mathbf{W}_{S_j}\left(\prod_{i=j+1}^{d}(\mathbf{W}_i)_{S_i}\right)$ is non-zero with probability 1. It follows that $\langle w, \left(\prod_{i=1}^{d}(\mathbf{W}_i)_{S_i}\right)_{*,j}\rangle$ is the inner product of a non-zero vector with a vector $w$ with continuous,

independent entries, and is thus non-zero with probability 1. By a union bound over all possible non-empty sets $S_j$, the desired result follows.

We now show that the second part of Assumption 2 holds. To do so, first let $g \sim \mathcal{N}(0, I_n)$. We demonstrate that $\mathbf{Pr}_{\mathbf{W}_1, \mathbf{w}_2, \dots, \mathbf{w}_d, g} [M(x) = 0] \leq 1 - \gamma\delta/100$. Here the entries of the $\mathbf{W}_i$'s are drawn independently but not necessarily identically from a continuous symmetric distribution. To see this, note that we can condition on the value of $g$, and condition at each step on the non-zero value of $y_i = \phi(W_{i+1}\phi(W_{i+2}\phi(\dots \phi(Ag)\dots)$. Then, over the randomness of $W_i$, note that the inner product of a row of $W_i$ and $y_i$ is strictly positive with probability at least $1/2$, and so each coordinate of $W_i y_i$ is strictly positive independently with probability $\geq 1/2$. It follows that $\phi(W_i y_i)$ is non-zero with probability at least $1 - 2^{-k_i}$. Thus

$$
\begin{aligned}
\mathbf{Pr}_{\mathbf{W}_1, \mathbf{w}_2, \dots, \mathbf{w}_d, g} [M(g) \neq 0] &\geq \prod_{i=1}^{d} (1 - 2^{-k_i}) \\
&\geq \left(1 - \frac{\delta\gamma}{16d}\right)^d \\
&\geq 1 - \delta\gamma/8
\end{aligned}
\tag{2}
$$

where the second inequality is by assumption. It follows by our first part that

$$
\mathbf{Pr}_{\mathbf{W}_1, \mathbf{w}_2, \dots, \mathbf{w}_d, g} [M(g) = 0] \leq \delta\gamma/8
$$

So by Markov's inequality,

$$
\mathbf{Pr}_{\mathbf{W}_1, \mathbf{w}_2, \dots, \mathbf{w}_d} [\mathbf{Pr}_g [M(g) = 0] \geq \gamma/8] \leq \delta
$$

Thus with probability $1 - \delta$ over the choice of $\mathbf{W}_1, \dots, \mathbf{W}_d$, we have $\mathbf{Pr}_g [M(g) = 0] \leq \gamma/8$ as desired. $\qquad\square$

**Lemma 3.2** *There is an algorithm which, given $g \sim \mathcal{N}(0, I_k)$, with probability $1 - \exp(-n^c)$ for any constant $c > 1$ (over the randomness in $g$), computes $\nabla M(g) \in \mathbb{R}^n$ with $O(n)$ queries to the network, and in $\mathrm{poly}(n)$ running time.*

*Proof.* Let $M_i(x) = \phi(\mathbf{W}_i\phi(\mathbf{W}_{i+1}\phi(\dots \phi(\mathbf{A}x))\dots))$ where $\phi$ is the ReLU. If $\nabla M(g)$ exists, there is an $\epsilon > 0$ such that $M$ is differentiable on $\mathcal{B}_\epsilon(g)$. We show that with good probability, if $g \sim \mathcal{N}(0, I_n)$ (or in fact, almost any continuous distribution), then $M(g)$ is continuous in the ball $\mathcal{B}_\epsilon(g) = \{x \in \mathbb{R}^n \mid \|x - g\|_2 < \epsilon\}$ for some $\epsilon$ which we will now compute.

First, we can condition on the event that $\|g\|_2^2 \leq (nd)^{10c}$, which occurs with probability at least $1 - \exp(-(nd)^{5c})$ by concentration results for $\chi^2$ distributions Laurent & Massart (2000). Now, fix any sign pattern $S_i \subseteq [k_i]$ for the $i$-th layer $M_i(x) = \phi(\mathbf{W}_i(\phi(\dots(\phi(\mathbf{A}x)\dots))$, and let $\mathcal{S} = (S_1, S_2, \dots, S_{d+1})$. We note that we can enforce the constraint that for an input $x \in \mathbb{R}^n$, the sign pattern of $M_i(x)$ is precisely $S_i$. To see this, note that after conditioning on a sign pattern for each layer, the entire network becomes linear. Thus each constraint that $\langle (\mathbf{W}_i)_{j,*}, M_{i+1}(x) \rangle \geq 0$ or $\langle (\mathbf{W}_i)_{j,*}, M_{i+1}(x) \rangle \leq 0$ can be enforced as a linear combination of the coordinates of $x$.

Now fix any layer $i \in [d+1]$ and neuron $j \in [k_i]$, WLOG $j \in S_i$. We now add the additional constraint that $\langle (\mathbf{W}_i)_{j,*}, M_{i+1}(x) \rangle \leq \eta$, where $\eta = \exp(-\mathrm{poly}(nd))$ is a value we will later choose. Thus, we obtain a linear program with $k + \sum_{i=1}^{d} k_i$ constraints and $n$ variables. The feasible polytope $\mathcal{P}$ represents the set of input points which satisfy the activation patterns $\mathcal{S}$ and are $\eta$-close to the discontinuity given by the $j$-th neuron in the $i$-th layer.

We can now introduce the following non-linear constraint on the input that $\|x\|_2 \leq (nd)^{10c}$. Let $\mathcal{B} = \mathcal{B}_{(nd)^{10c}}(\vec{0})$ be the feasible region of this last constraint, and let $\mathcal{P}^* = \mathcal{P} \cap \mathcal{B}$. We now bound the Lesbegue measure (volume) $V(\mathcal{P}^*)$ of the region $\mathcal{P}$. First note that $V(\mathcal{P}^*) \leq V(\mathcal{P}')$, where $V(\mathcal{P}')$ is the region defined by the set of points which satisfy:

$$
\begin{aligned}
\langle y, x \rangle &\geq 0 \\
\langle y, x \rangle &\leq \eta \\
\|x\|_2^2 &\leq (nd)^{10c}
\end{aligned}
\tag{3}
$$

where each coordinate of the vector $y \in \mathbb{R}^n$ is a linear combination of products of the weight matrices $\mathbf{W}_\ell$, $\ell \geq i$. One can see that the first two constraints for $\mathcal{P}'$ are also constraints for $\mathcal{P}^*$, and the last constraint is precisely $\mathcal{B}$, thus $\mathcal{P}^* \subset \mathcal{P}'$ which completes the claim of the measure of the latter being larger. Now we can rotate $\mathcal{P}'$ by the rotation

which sends $y \to \|y\|_2 \cdot e_1 \in \mathbb{R}^n$ without changing the volume of the feasible region. The resulting region is contained in the region $\mathcal{P}''$ given by

$$0 \leq x_1 \leq \eta \|x\|_\infty^2 \leq (nd)^{10c} \tag{4}$$

Finally, note that $\mathcal{P}'' \subset \mathbb{R}^n$ is a Eucledian box with $n - 1$ side lengths equal to $(nd)^{10c}$ and one side length of $\|y\|_2 \eta$, and thus $V(\mathcal{P}'') \leq \|y\|_2 \eta (nd)^{10nc}$. Now note we can assume that the entries of the weight matrices $\mathbf{A}, \mathbf{W}_1, \ldots, \mathbf{W}_d$ are specified in polynomially many (in $n$) bits, as if this were not the case the output $M(x)$ of the network would not have polynomial bit complexity, and could not even be read in poly$(n)$ time. Equivalently, we can assume that our running time is allowed to be polynomial in the number of bits in any value of $M(x)$, since this is the size of the input to the problem. Given this, since the coordinates of $y$ were linear combinations of products of the coordinates of the weight matrices, and note that each of which is at most $2^{n^C}$ for some constant $C$ (since the matrices have polynomial bit-complexity), we have that $\mathcal{P}^* \leq \eta 2^{n^C} (nd)^{10nc}$ as needed.

Now the pdf of a multi-variate Gaussian is upper bounded by 1, so $\mathbf{Pr}_{g \sim \mathcal{N}(0,I_n)}[g \in \mathcal{P}^*] \leq V(\mathcal{P}^*) \leq \eta 2^{n^C} (nd)^{10nc}$. It follows that the probability that a multivariate Gaussian $g \sim \mathcal{N}(0, I_n)$ satisfies the sign pattern $\mathcal{S}$ and is $\eta$ close to the boundary for the $j$-th neuron in the $i$-th layer. Now since there are at most $2^k \cdot \prod_{i=1}^d 2^{k_i} \leq 2^{nd}$ possible combinations of sign patterns $\mathcal{S}$, it follows that the the probability that a multivariate Gaussian $g \in \mathcal{N}(0, I_n)$ is $\eta$ close to the boundary for the $j$-th neuron in the $i$-th layer is at most $\eta 2^{n^C} (nd)^{10nc} 2^{nd}$. Union bounding over each of the $k_i$ neurons in layer $i$, and then each of the $d$ layers, it follows that $g \in \mathcal{N}(0, I_n)$ is $\eta$ close to the boundary for any discontinuity in $M(x)$ is at most $\eta 2^{n^C} (nd)^{10nc+1} 2^{nd}$. Setting $\eta \leq 2^{(nd)^{20c+1}} 2^{-n^C}$, it follows that with probability at least $1 - \exp(-(nd)^c)$, the network evaluated at $g \in \mathcal{N}(0, I_n)$ is at least $\eta$ close from all boundaries (note that $C$ is known to us by assumption).

Now we must show that perturbing the point $g$ by any vector with norm at most $\epsilon$ results in a new point $g'$ which still has not hit one of the boundaries. Note that $M(g)$ is linear in an open ball around $g$, so the change that can occur in any intermediate neuron after perturbing $g$ by some $v \in \mathbb{R}^n$ is at most $\|\mathbf{A}\|_2 \prod_{i=1}^d \|\mathbf{W}_i\|_2$, where $\|\cdot\|_2$ is the spectral norm. Now since each entry in the weight matrix can be specified in polynomially many bits, the Frobenius norm of each matrix (and therefore the spectral norm), is bounded by $n^2 2^{n^C}$ for some constant $C$. Thus

$$\|\mathbf{A}\|_2 \prod_{i=1}^d \|\mathbf{W}_i\|_2 \leq (n^2 2^{n^C})^{d+1} = \beta$$

and setting $\epsilon = \eta/\beta$, it follows that $M(x)$ is differentiable in the ball $\mathcal{B}_\epsilon(x)$ as needed.

We now generate $u_1, u_2, \ldots, u_n \sim \mathcal{N}(0, I_n)$, which are linearly independent almost surely. We set $v_i = \frac{\epsilon u_i}{2\|u_i\|_2}$. Since $M(g)$ is a ReLU network which is differentiable on $\mathcal{B}_\epsilon(g)$, it follows that $M(g)$ is a linear function on $\mathcal{B}_\epsilon(g)$, and moreover $v_i \in \mathcal{B}_\epsilon(g)$ for each $i \in [n]$. Thus for any $c < 1$ we have $\frac{M(g) - M(g + cv_i)}{c} = \nabla_{v_i} M(x)$, thus we can compute $\nabla_{v_i} M(x)$ for each $i \in [n]$. Finally, since the directional derivative is given by $\nabla_{v_i} M(x) = \langle \nabla M(x), v_i/\|v_i\|_2 \rangle$, and since $v_1, \ldots, v_n$ are linearly independent, we can set up a linear system to solve for $\nabla M(x)$ exactly in polynomial time, which completes the proof. $\square$

**Lemma 3.3** *Let* $\mathbf{V} \in \mathbb{R}^{r \times k}$ *be a fixed at most rank* $ck$ *matrix for* $c \leq 1/2$. *Then the number of sign patterns* $S \subset [k]$ *with at most* $k/2$ *non-zeros spanned by the rows of* $\mathbf{V}$ *is at most* $\frac{2^k}{\sqrt{k}}$. *In other words, the set* $\mathcal{S}(\mathbf{V}) = \{sign(w) \mid w \in span(\mathbf{V}), nnz(w) \leq \frac{k}{2}\}$ *has size at most* $\frac{2^k}{\sqrt{k}}$.

*Proof.* Any vector $w$ in the span of the rows of $\mathbf{V}$ can be expressed as a linear combination of at most $ck$ rows of $\mathbf{V}$. So create a variable $x_i$ for each coefficient $i \in [ck]$ in this linear combination, and let $f_j(x)$ be the linear function of the $x_i's$ which gives the value of the $j$-th coordinate of $w$. Then $f(x) = (f_1(x), \ldots, f_k(x))$ is a $k$-tuple of polynomials, each in $ck$-variables, where each polynomial has degree 1. By Theorem 4.1 of Hall et al. (2010), it follows that the number of sign patterns which contain at most $k/2$ non-zero entries is at most $\binom{ck+k/2}{ck}$. Setting $c \leq 1/2$, this is at most $\binom{k}{k/2} \leq \frac{2^k}{\sqrt{k}}$. $\square$

**Theorem 3.4** *Suppose the network* $M(x) = w^T \phi(W_1 \phi(W_2 \phi(\ldots W_d \phi(Ax)) \ldots))$, *where* $\phi$ *is the ReLU, satisfies* **Assumption 1 and 2**. *Then the algorithm given in Figure 1 makes* $O(kn \log(k/\delta)/\gamma)$ *queries to* $M(x)$ *and returns in* $poly(n, k, 1/\gamma, \log(1/\delta))$-*time a subspace* $V \subseteq span(A)$ *of dimension at least* $\frac{k}{2}$ *with probability* $1 - \delta$.

*Proof.* First note that by Lemma 3.2, we can efficiently compute each gradient $\nabla M(g^i)$ using $O(n)$ queries to the network. After querying for the gradient $z^i = \nabla M(g^i)$ for $r' \le r$ independent Gaussian vectors $g^i \in \mathbb{R}^n$, we obtain the vector of gradients $\mathbf{Z} = \mathbf{V} \cdot \mathbf{A} \in \mathbb{R}^{r' \times k}$. Now suppose that $\mathbf{V}$ had rank $ck$ for some $c \le 1/2$. Now consider the gradient $\nabla(g^{r'+1})$, which can be written as $z^{r'+1} = w^T \left( \prod_{i=1}^d (\mathbf{W}_i)_{S_i^{r'+1}} \right) \text{Diag}(\text{sign}(\mathbf{A}g^{r'+1}))\mathbf{A}$. Thus we can write $z^{r'+1} = \mathbf{V}_{r'+1}\mathbf{A}$, where $\mathbf{V}_{r'+1} \in \mathbb{R}^k$ is a row vector which will be appended to the matrix $\mathbf{V}$ to form a new $\mathbf{Z} = \mathbf{VA} \in \mathbb{R}^{r'+1 \times k}$ after the $(r'+1)$-th gradient is computed. Specifically, for any $j \in [r]$, we have: $\mathbf{V}_j = w^T \left( \prod_{i=1}^d (\mathbf{W}_i)_{S_i^j} \right) \text{Diag}(\text{sign}(\mathbf{A}g^j))$.

Let $\mathcal{E}_{r'+1}$ be the event that $M(g^{r'+1}) \ne 0$. It follows necessarily that, conditioned on $\mathcal{E}_{r'+1}$, we have that $\text{sign}(\mathbf{V}_{r'+1}) = S_0^{r'+1}$. The reason is as follows: if $M(g^{r'+1}) \ne 0$, then we could not have $S_j^{r'+1} = \emptyset$ for any $j \in \{0, 1, 2, \ldots, d\}$, since this would result in $M(g^{r'+1}) = 0$. It follows that the conditions for Assumption 2 to apply hold, and we have that $w^T \left( \prod_{i=1}^d (\mathbf{W}_i)_{S_i^{r'+1}} \right) \in \mathbb{R}^k$ is entry-wise non-zero. Given this, it follows that the sign pattern of

$$\mathbf{V}_{r'+1} = w^T \left( \prod_{i=1}^d (\mathbf{W}_i)_{S_i^{r'+1}} \right) \text{Diag}(\text{sign}(\mathbf{A}g^j))$$

will be precisely $S_0^{r'+1}$, as needed.

Now by Lemma 3.3, the number of sign patterns of $k$-dimensional vectors with at most $k/2$ non-zero entries which are contained in the span of $\mathbf{V}$ is at most $\frac{2^k}{\sqrt{k}}$. Let $\mathcal{S}$ be the set of sign patterns with at most $k/2$ non-zeros and such that for every $S \in \mathcal{S}$, $S$ is not a sign pattern which can be realized in the row span of $\mathbf{V}$. It follows that $|\mathcal{S}| \ge \frac{1}{2} 2^k - \frac{2^k}{\sqrt{k}} \ge \frac{1}{4} 2^k$, so by Assumption 1, we have that $\Pr\left[ \text{sign}(Ag^{r'+1}) \in \mathcal{S} \right] \ge \gamma/4$. By a union bound, we have $\Pr\left[ \text{sign}(Ag^{r'+1}) \in \mathcal{S}, \mathcal{E}_{r'+1} \right] \ge \gamma/8$

Conditioned on $\text{sign}(Ag^{r'+1}) \in \mathcal{S})$ and $\mathcal{E}_{r'+1}$ simultaneously, it follows that adding the row vector $\mathbf{V}_{r'+1}$ to the matrix $\mathbf{V}$ will increase its rank by 1. Thus after $O(\log(k/\delta)/\gamma)$ repetitions, the rank of $\mathbf{V}$ will be increased by at least 1 with probability $1 - \delta/k$. By a union bound, after after $r = O(k\log(k/\delta)/\gamma)$, $\mathbf{V} \in \mathbb{R}^{r \times k}$ will have rank at least $k/2$ with probability at least $1 - \delta$, which implies that the same will hold for $\mathbf{Z}$ since $\text{rank}(\mathbf{Z}) \ge \text{rank}(\mathbf{V})$, which is the desired result. $\qquad\square$

# B   MISSING PROOFS FROM SECTION 4

**Proposition 4.1**   *Let $V \subset \mathbb{R}^n$ be a subspace of dimension $k' < k$, and fix any $\epsilon_0 > 0$. Then we can find a vector $x$ with*

$$0 \le \sigma_V(x) - 1 \le 2\epsilon_0$$

*in expected $O(1/\gamma + N\log(1/\epsilon_0))$ time. Moreover, with probability $\gamma/2$ we have that $\nabla_x \sigma_V(x) > \eta/4$ and the tighter bound of*

$$\epsilon_0 \eta 2^{-N} \le \sigma_V(x) - 1 \le 2\epsilon_0$$

*Proof.* We begin by generating Gaussians $g_1, \ldots$ and computing $M_V(g_i)$. By Property 3 of the network assumptions, we need only repeat the process $1/\gamma$ times until we obtain an input $y = (\mathbb{I}_n - \mathbf{P}_V)g_i$ with $M(y) = M_V(g_i) = 1$. Since all activation functions satisfy $\phi_i(0) = 0$, we know that $\sigma_V(0 \cdot g_i) = 0$, and $\sigma_V(g_i) = 1$. Since $\sigma$ is continuous, it follows that $\psi_{g_i}(c) = \sigma_V(c \cdot g_i)$ is a continuous function $\psi : \mathbb{R} \mapsto \mathbb{R}$. By the intermediate value theorem, there exists a $c^* \le 1$ such that $\sigma_V(c^*g_i) = 1$. We argue we can find a $c$ with $|c - c^*| \le \epsilon_0 2^{-N}$ in time $O(N\log(1/\epsilon_0))$.

To find $c$, we can perform a binary search. We first try $c_0 = 1/2$, and if $\psi_{g_i}(c_0) = 0$, we recurse into $[1/2, 1]$, otherwise if $\psi_{g_i}(c_0) = 1$ we recurse into $[0, 1/2]$. Thus, we always recurse into an interval where $\psi_{g_i}$ switches values. It follows that we can find a $c$ with $|c - c^*| \le \epsilon_0 \|g_i\|_2 2^{-N}$ in time $O(N\log(\|g_i\|_2/\epsilon_0))$ for some $c^*$ with $\sigma_V(c^*g_i) = 1$. Now observe that it suffices to binary search a total of $O(N\log(\|g_i\|_2/\epsilon_0))$ times, since $2^N$ is an upper bound on the Lipschitz constant of $\psi_x$, which gives $0 \le \sigma_V(cg_i) - 1 \le \epsilon_0$. Now the expected running time to do this is $O(1/\gamma + N\log(\|g_i\|_2/\epsilon_0))$, but since $\|g_i\|_2$ has Gaussian tails, the expectation of the maximum value of $\|g_i\|_2$ over

$1/\gamma$ repetitions is $O(\log(1/\gamma)\sqrt{n})$, and thus the expected running time reduces to the stated bound, which completes the first claim of the Proposition.

For the second claim, note that $\nabla_{g_i}\sigma_V(c^*g_i) > 0$ by construction of the binary search, and since $\sigma_V(c^*g_i) > 0 = 1$, by Property 4 with probability $\gamma$ we have that $\nabla_{g_i}\sigma_V(g_i) > \eta$. Now with probability $1 - \gamma/2$, we have that $\|g_i\|_2^2 \leq O(n\log(1/\gamma))$ (see Lemma 1 Laurent & Massart (2000)), so by a union bound both of these occur with probability $\gamma/2$. Now since $\|(c^* - c)x\|_2 \leq \epsilon_0 2^{-N}$ (after rescaling $N$ by a factor of $\log(\|g_i\|_2) = O(\log(n))$), and since $2^N$ is also an upper bound on the spectral norm of the Hessian of $\sigma$ by construction, it follows that $\nabla_{g_i}\sigma_V(cg_i) > \eta/2$.

Now we set $x \leftarrow cg_i + c\epsilon_0 2^{-N}g_i/(\|cg_i\|_2)$. First note that this increases $\sigma_V(cx) - 1$ by at most $\epsilon_0$, so $\sigma(cx) - 1 \leq 2\epsilon_0$, so this does not affect the first claim of the Proposition. But in addition, note that conditioned on the event in the prior paragraph, we now have that $\sigma_V(x) > 1 + \eta\epsilon_0 2^{-N}$. The above facts can be seen by the fact that $2^N$ is polynomially larger than the spectral norm of the Hessian of $\sigma$, thus perturbing $x$ by $\epsilon_0 2^{-N}$ additive in the direction of $x$ will result in a positive change of at least $\frac{1}{2}(\eta/4)(\epsilon_0 2^{-N})$ in $\sigma$. Moreover, by applying a similar argument as in the last paragraph, we will have $\nabla_x\sigma_V(cx) > \eta/4$ still after this update to $x$. $\qquad\square$

**Lemma 4.2** *Fix any $\epsilon, \delta > 0$, and let $N = poly(n, k, \frac{1}{\gamma}, \sum_{i=1}^d \log(L_i), \sum_{i=1}^d \log(\kappa_i), \log(\frac{1}{\eta}), \log(\frac{1}{\epsilon}), \log(\frac{1}{\delta}))$. Then given any subspace $V \subset \mathbb{R}^n$ with dimension $dim(V) < k$, and given $x \in \mathbb{R}^n$, such that $\epsilon_0\eta 2^{-N} \leq \sigma_V(x) - 1 \leq 2\epsilon_0$ where $\epsilon_0 = \Theta(2^{-N^C}/\epsilon)$ for a sufficiently large constant $C = O(1)$, and $\nabla_x\sigma_V(x) > \eta/2$, then with probability $1 - 2^{-N/n^2}$, we can find a vector $v \in \mathbb{R}^n$ in expected $poly(N)$ time, such that*

$$\|\mathbf{P}_{Span(\mathbf{A})}v\|_2 \geq (1 - \epsilon)\|v\|_2$$

*and such that $\|\mathbf{P}_V v\|_2 \leq \epsilon\|v\|_2$.*

*Proof.* We generate $g_1, g_2, \ldots, g_n \sim \mathcal{N}(0, I_n)$, and set $u_i = g_i 2^{-N} - x/\|x\|_2$. We first condition on the event that $\|g_i\|_2 \leq N$ for all $i$, which occurs with probability $1 - 2^{-N/n^2}$. Note $\nabla_{u_i}[\sigma_V(x)] = w^T\mathbf{A}(\mathbb{I}_n - \mathbf{P}_V)u_i$, where $w^T = \nabla[\sigma(\mathbf{A}(\mathbb{I}_n - \mathbf{P}_V)x)]^T \in \mathbb{R}^k$, which does not depend on $u_i$. Thus $w^T\mathbf{A}(\mathbb{I}_n - \mathbf{P}_V)$ is a vector in the row span of $\mathbf{A}(\mathbb{I}_n - \mathbf{P}_V)$. We can write

$$M_V(x + cu_i) = \tau\left((\sigma_V(x) + cw^T\mathbf{A}(\mathbb{I}_n - \mathbf{P}_V)u_i + \Xi(cu_i)\right)$$

where $\Xi(cu_i) = O(\|c(\mathbb{I}_n - \mathbf{P}_V)u_i\|_2^2 2^N) = O(\|cu_i\|_2^2 2^N)$ is the error term for the linear approximation. Note that the factor of $N$ comes from the fact that the spectral norm of the Hessian of $\sigma : \mathbb{R}^n \to \mathbb{R}$ can be bounded by $\prod_i \kappa_i L_i \leq 2^N$. Fix some $\beta > 0$. We can now binary search again, as in Proposition 4.1, with $O(\log(N/\beta))$ iterations over $c$, querying values $M_V(x + cu_i)$ to find a value $c = c_i > 0$ such that

$$1 - \beta \leq M_V(x + cu_i) \leq 1$$

so

$$\mathbf{1} - \beta \leq (\sigma_V(x) + c_i w^T\mathbf{A}(\mathbb{I}_n - \mathbf{P}_V)u_i + \Xi(c_iu_i)) \leq 1$$

We first claim that the $c_i$ which achieves this value satisfies $\|c_iu_i\|_2 \leq (10 \cdot 2^{-N}\epsilon_0/\eta)$. To see this, first note that by Proposition 4.1, we have $\nabla_x\sigma_V(x) > \eta/4$ with probability $\gamma$. We will condition on this occurring, and if it fails to occur we argue that we can detect this and regenerate $x$. Now conditioned on the above, we first claim that $\nabla_{u_i}\sigma_V(x) \geq \eta/8$, which follows from the fact that we can bound the angle between the unit vectors in the directions of $u_i$ and $x$ by

$$\cos(angle(u_i, x)) = \left\langle \frac{u_i}{\|u\|_2}, \frac{x}{\|x\|_2} \right\rangle \geq (1 - n/2^{-N}) > (1 - \eta/2^{-N/2})$$

along with the fact that we have $\nabla_x\sigma_V(x) > \eta/4$. Since $|\sigma_V(x) - 1| < 2\epsilon_0 < 2^{-N^C}$, and since $2^N$ is an upper bound on the spectral norm of the Hessian of $\sigma$, we have that $\nabla_{u_i}\sigma_V(x + cu_i) > \eta/8 + 2^{-N} > \eta/10$ for all $c < 2^{-2N}$. In other words, if $H$ is the hessian of $\sigma$, then perturbing $x$ by a point with norm $O(c) \leq 2^{-2N}$ can change the value of the gradient by a vector of norm at most $2^{2N}\|H\|_2 \leq 2^{-N}$, where $\|H\|_2$ is the spectral norm of the Hessian. It follows that setting $c = (10 \cdot 2^{-N}\epsilon_0/\eta)$ is sufficient for $\sigma_V(x + cu_i) < 1$, which completes the above claim.

Now observe that if after binary searching, the property that $c \leq (10 \cdot 2^{-N}\epsilon_0/\eta)$ does not hold, then this implies that we did not have $\nabla\sigma(x) > \eta/4$ to begin with, so we can throw away this $x$ and repeat until this condition does hold. By Assumption 4, we must only repeat $O(1/\gamma)$ times in expectation in order for the assumption to hold.

Next, also note that we can bound $c_i \geq \epsilon_0\eta 2^{-N}/N$, since $2^N$ again is an upper bound on the norm of the gradient of $\sigma$ and we know that $\sigma_V(x) - 1 > \epsilon_0\eta 2^{-N}$. Altogether, we now have that $|\Xi(c_iu_i)| \leq c_i^2 2^N \leq (10 \cdot 2^{-N}\epsilon_0/\eta)^2 2^N$.

We can repeat this binary search to find $c_i$ for $n$ different perturbations $u_1, \ldots, u_n$, and obtain the resulting $c_1, \ldots, c_n$, such that for each $i \in [n]$ we have

$$1 - \sigma_V(x) - \Xi(c_i u_i) - \beta_i = c_i w^T \mathbf{A}(\mathbb{I}_n - \mathbf{P}_V) u_i$$

where $\beta_i$ is the error obtained from the binary seach on $c_i$, and therefore satisfies $|\beta_i| \leq 2^{-N^2} \epsilon_0^2$ taking $\text{poly}(N)$ iterations in the search. Now we know $c_i$ and $u_i$, so we can set up a linear system

$$\min_y \|y^T \mathbf{B} - b^T\|_2^2$$

for an unknown $y \in \mathbb{R}^k$ where the $i$-th column of $\mathbf{B}$ is given by $\mathbf{B}_i = u_i \in \mathbb{R}^n$, and $b_i = 1/c_i$ for each $i \in [n]$. First note that the set $\{u_1, u_2, \ldots, u_n\}$ is linearly independent with probability 1, since $\{u_1 - x, u_2 - x, \ldots, u_n - x\}$ is just a set of Gaussian vectors. Thus $\mathbf{B}$ has rank $n$, and the above system has a unique solution $y^*$.

Next, observe that setting $\widehat{y} = \frac{w^T \mathbf{A}(\mathbb{I}_n - \mathbf{P}_V)}{(1 - \sigma_V(x))}$, we obtain that for each $i \in [n]$:

$$(\widehat{y}B)_i = \frac{1}{c_i} - \frac{1}{c_i} \frac{\Xi(c_i u_i) - \beta_i}{1 - \sigma_V(x)}$$

Thus

$$
\begin{aligned}
\|\widehat{y}^T B - b^T\|_2 &\leq \left( \sum_{i=1}^n \left( \frac{1}{c_i} \cdot \frac{\Xi(c_i u_i) - \beta_i}{1 - \sigma_V(x)} \right)^2 \right)^{1/2} \\
&\leq \left( n \left( \frac{1}{c_i} \cdot \frac{(10 \cdot 2^{-N} \epsilon_0/\eta)^2 2^N - 2^{-N^2} \epsilon_0^2}{1 - \sigma_V(x)} \right)^2 \right)^{1/2} \qquad (5) \\
&\leq (2^{-N})^{O(1)} \frac{\epsilon_0^2}{c_i(1 - \sigma(x))}
\end{aligned}
$$

Now setting $y^*$ such that $(y^*)^T \mathbf{B} = b^T$, we have that cost of the optimal solution is 0, and $\|\widehat{y}\mathbf{B} - b^T\|_2 \leq (2^{-N})^{O(1)} \frac{\epsilon_0^2}{c_i(1-\sigma(x))}$, so $\|(y^* - \widehat{y})\mathbf{B}\|_2 \leq (2^{-N})^{O(1)} \frac{\epsilon_0^2}{c_i(1-\sigma(x))}$. By definition of the minimum singular value of $B$, it follows that $\|y^* - \widehat{y}\|_2 \leq \frac{1}{\sigma_{\min}(\mathbf{B})} (2^{-N})^{O(1)} \frac{\epsilon_0^2}{c_i(1-\sigma(x))}$. Now using the fact that $\mathbf{B} = 2^{-N} \mathbf{G} + \mathbf{X}$ where $\mathbf{G}$ is a Gaussian matrix and $\mathbf{X}$ is the matrix with each row equal to $x$, we can apply Theorem 1.6 of Cook et al. (2018), we have $\sigma_{\min}(\mathbf{B}) \geq 1/(2^{-N})^{O(1)}$, So $\|y^* - \widehat{y}\|_2 \leq (2^{-N})^{O(1)} \frac{\epsilon_0^2}{c_i(1-\sigma(x))}$. Thus we have $\|y^*\|_2 \leq \|\widehat{y}\|_2 + (2^{-N})^{O(1)} \frac{\epsilon_0^2}{c_i(1-\sigma(x))}$, and moreover note that $\|\widehat{y}\|_2 \geq \|\nabla \sigma(x)\|_2 \frac{1}{1-\sigma(x)} > \frac{\eta}{8(1-\sigma(x))}$. So we have that

$$
\begin{aligned}
\frac{\|y^* - \widehat{y}\|_2}{\|\widehat{y}\|_2} &\leq \frac{(2^{-N})^{O(1)} \frac{\epsilon_0^2}{c_i(1-\sigma(x))}}{\|y^*\|_2} \\
&\leq (2^{-N})^{O(1)} \frac{\epsilon_0^2}{c_i} \\
&\leq (2^{-N})^{O(1)} \epsilon_0 \qquad (6) \\
&\leq (2^{-N})^{O(1)} \epsilon_0 \\
&\leq 2^{-N/2}
\end{aligned}
$$

Where the last inequality holds by taking $C$ larger than some constant in the definition of $\epsilon_0$. Thus $\|y^* - \widehat{y}\|_2 \leq 2^{-N/2} \|\widehat{y}\|_2$, thus $\|y^* - \widehat{y}\|_2 \leq 2 \cdot 2^{-N/2} \|y^*\|_2 \leq \epsilon \|y^*\|_2$ after scaling $N$ up by a factor of $\log^2(1/\epsilon)$. Thus by setting $v = y^*$, and observing that $\widehat{y}$ is in the span of $\mathbf{A}$, we ensure $\|\mathbf{P}_{\text{Span}(\mathbf{A})} v\|_2 \geq (1 - \epsilon) \|v\|_2$ as desired. For the final claim, note that if we had $y^* = \widehat{y}$ exactly, we would have $\|\mathbf{P}_V y^*\|_2 = 0$, since $\widehat{y}$ is orthogonal to the subspace $V$. It follows that since $\|y^* - \widehat{y}\|_2^2 \leq \epsilon^2 \|y^*\|_2^2$, we have $\|(\mathbb{I}_n - \mathbf{P}_V) y^*\|_2^2 \geq (1 - \epsilon^2) \|y^*\|_2^2$, so by the Pythagorean theorem, we have $\|P_V y^*\|_2^2 = \|y^*\|_2^2 - \|(\mathbb{I}_n - \mathbf{P}_V) y^*\|_2^2 \leq \epsilon^2 \|y^*\|_2^2$ as desired. $\qquad \square$

**Theorem 4.3** *Suppose the network $M(x) = \tau(\sigma(\mathbf{A}x))$ satisfies the conditions described at the beginning of this section. Then Algorithm 2 runs in $\text{poly}(N)$ time, making at most $\text{poly}(N)$ queries to $M(x)$, where $N =$*

$poly(n, k, \frac{1}{\gamma}, \sum_{i=1}^{d} \log(L_i), \sum_{i=1}^{d} \log(\kappa_i), \log(\frac{1}{\eta}), \log(\frac{1}{\epsilon}), \log(\frac{1}{\delta}))$, *and returns with probability* $1 - \delta$ *a subspace* $V \subset \mathbb{R}^n$ *of dimension* $k$ *such that for any* $x \in V$, *we have*

$$\|\mathbf{P}_{Span(\mathbf{A})}x\|_2 \geq (1 - \epsilon)\|x\|_2$$

*Proof.* We iteratively apply Lemma 4.2, each time appending the output $v \in \mathbb{R}^n$ of the proposition to the subspace $V \subset \mathbb{R}^n$ constructed so far. WLOG we can assume $v$ is a unit vector by scaling it. Note that we have the property at any given point in time $k' < k$ that $V = \text{Span}(v_1, \ldots, v_{k'})$ where each $v_i$ satisfies that $\|\mathbf{P}_{\text{Span}\{v_1, \ldots, v_{i-1}\}} v_i\|_2 \leq \epsilon$. Note that the latter fact implies that $v_1, \ldots v_{k'}$ are linearly independent. Thus at the end, we recover a rank $k$ subspace $V = \text{Span}(v_1, \ldots, v_k)$, with the property that $\|\mathbf{P}_{\text{Span}(\mathbf{A})} v_i\|_2^2 \geq (1 - \epsilon)\|v_i\|_2^2$ for each $i \in [k]$.

Now let $\mathbf{V} \in \mathbb{R}^{n \times n}$ be the matrix with $i$-th column equal to $v_i$. Fix any unit vector $x = \mathbf{V}a \in V$, where $a \in \mathbb{R}^n$ is uniquely determined by $x$. Let $\mathbf{V} = \mathbf{V}^+ + \mathbf{V}^-$ where $\mathbf{V}^+ = \mathbf{P}_{\text{Span}(A)}\mathbf{V}$ and $\mathbf{V}^- = \mathbf{V} - \mathbf{V}^+$ Then $x = \mathbf{V}^+ a + \mathbf{V}^- a$, and

$$
\begin{aligned}
\|(\mathbb{I}_n - \mathbf{P}_{\text{Span}(\mathbf{A})})x\|_2 &\leq \|(\mathbb{I}_n - \mathbf{P}_{\text{Span}(\mathbf{A})})\mathbf{V}^+ a\|_2 + \|(\mathbb{I}_n - \mathbf{P}_{\text{Span}(\mathbf{A})})\mathbf{V}^- a\|_2 \\
&\leq \|(\mathbb{I}_n - \mathbf{P}_{\text{Span}(\mathbf{A})})\mathbf{V}^- a\|_2 \\
&\leq \|\mathbf{V}^-\|_2 \|a\|_2
\end{aligned}
\tag{7}
$$

First note that by the construction of the $v_i$'s, each column of $\mathbf{V}^-$ has norm $O(\epsilon)$, thus $\|\mathbf{V}^-\|_2 \leq O(\sqrt{n}\epsilon)$. Moreover, since $\|x\|_2 = 1$, it follows that $\|a\|_2 \leq \frac{1}{\sigma_{\min}(\mathbf{V})}$, which we now bound. Since $\|\mathbf{P}_{\text{Span}\{v_1, \ldots, v_{i-1}\}} v_i\|_2 \leq \epsilon$ for each $i \in [k]$, we have

$$
\begin{aligned}
\|\mathbf{V}a\|_2 &\geq \|\sum_{i=1}^{n} v_i a_i\|_2 \\
&\geq \|\sum_{i=1}^{n}(\mathbb{I}_n - \mathbf{P}_{\text{Span}\{v_1, \ldots, v_{i-1}\}})v_i a_i + \sum_{i=1}^{n}\mathbf{P}_{\text{Span}\{v_1, \ldots, v_{i-1}\}} v_i a_i\|_2 \\
&\geq \|\sum_{i=1}^{n}(\mathbb{I}_n - \mathbf{P}_{\text{Span}\{v_1, \ldots, v_{i-1}\}})v_i a_i\|_2 - \|\sum_{i=1}^{n}\mathbf{P}_{\text{Span}\{v_1, \ldots, v_{i-1}\}} v_i a_i\|_2 \\
&\geq \|\sum_{i=1}^{n}(\mathbb{I}_n - \mathbf{P}_{\text{Span}\{v_1, \ldots, v_{i-1}\}})v_i a_i\|_2 - O(\epsilon)\|a\|_2 \\
&= \left(\|\sum_{i=1}^{n}(\mathbb{I}_n - \mathbf{P}_{\text{Span}\{v_1, \ldots, v_{i-1}\}})v_i a_i\|_2^2 - O(\epsilon)\|a\|_2\|\sum_{i=1}^{n}(\mathbb{I}_n - \mathbf{P}_{\text{Span}\{v_1, \ldots, v_{i-1}\}})v_i a_i\|_2 + O(\epsilon^2)\|a\|_2^2\right)^{1/2} \\
&= \left(\|\sum_{i=1}^{n}(\mathbb{I}_n - \mathbf{P}_{\text{Span}\{v_1, \ldots, v_{i-1}\}})v_i a_i\|_2^2 - O(\epsilon)\|a\|_2^2\right)^{1/2} \\
&= \left(\sum_{i=1}^{n}\|(\mathbb{I}_n - \mathbf{P}_{\text{Span}\{v_1, \ldots, v_{i-1}\}})v_i a_i\|_2^2 - O(\epsilon)\|a\|_2^2\right)^{1/2} \\
&\geq \left(\sum_{i=1}^{n}(1 - O(\epsilon^2))|a_i|^2 - O(\epsilon)\|a\|_2^2\right)^{1/2} \\
&\geq \left(\|a\|_2^2 - O(\epsilon)\|a\|_2^2\right)^{1/2} \\
&\geq (1 - O(\epsilon))\|a\|_2
\end{aligned}
\tag{8}
$$

Thus $\sigma_{\min}(\mathbf{V}) \geq (1 - O(\epsilon))$, so we have $\|(\mathbb{I}_n - \mathbf{P}_{\text{Span}(\mathbf{A})})x\|_2 \leq \|\mathbf{V}^-\|_2 \frac{1}{\sigma_{\min}(\mathbf{V})} \leq 2\sqrt{n}\epsilon$. By the Pythagorean theorem: $\|\mathbf{P}_{\text{Span}(\mathbf{A})}x\|_2^2 = 1 - \|(\mathbb{I}_n - \mathbf{P}_{\text{Span}(\mathbf{A})})x\|_2^2 \geq 1 - O(n\epsilon^2)$. Thus we can scale $\epsilon$ by a factor of $\Theta(1/\sqrt{n})$ in the call to Lemma 4.2, which gives the desired result of $\|\mathbf{P}_{\text{Span}(\mathbf{A})}x\|_2 \geq 1 - \epsilon$. $\qquad\square$

