# OpenReview forum: "Span Recovery for Deep Neural Networks with Applications to Input Obfuscation"
_ICLR.cc/2020/Conference — Accept (Poster)_

### Official Review · AnonReviewer3 · 2019-10-24
**Official Blind Review #3**

**Rating:** 6

**Review:**

** Summary
The paper studies how to recover the span of a NN from a limited number of queries. The problem belongs to the general question of how to reconstruct functions from black-box interaction and it may find application in obfuscation attacks where very large perturbations of the input do not affect the output. The main contribution of the paper is on the theoretical analysis of a simple non-adaptive and a more sophisticated adaptive algorithm. The main finding is that under mild conditions on the structure of the NN partial recovery is possible. The empirical validation show that in practice, it is often the case the full span recover is actually possible, as the structure and weights of common NN are "friendly" enough.

** Evaluation
While the content of the paper lies a bit off my expertise, my impression is that this is a solid technical and theoretical contribution.

Detailed comments:
1- Theoretical results: The properties proved in Thm.3.4 and 4.3 are quite powerful, showing that (partial/approximate) span recovery is possible with a relatively small amount of samples (of order of n*k, where n is the original input size and k is the size of the span), in a computationally efficient way (in particular for the non-adaptive algorithm), and for Relu NN or NN with differentiable layers and final threshold function. The main question is of course the validity of the assumptions needed to prove the theorems. Asm.1 and 2 are overall reasonable and they are very well supported by Lemma 3.1. The first two assumptions in page 6 are straightforward, while I'll less convinced of 3 and 4. In fact, they need to hold for any subspace V of any dimension smaller than k. I wonder whether the assumptions may become less and less likely as the size of the subspace decrease.
2- The theorems and the paper are mostly well written but some parts may be clearer.
3- Alg.1: the computation of the gradient is never really explained apart from the high-level lemma 3.2. While an actual algorithm is reported in the appendix, it would be better to have it explained already in the main text.
4- Right after Lemma 3.2 it is said "which demonstrates the claim". I am not sure which claim it refers to.
5- In alg.1 there is a parameter r which defines the number of queries of the algorithm. Thm 3.4 provides an upper bound on the number of queries needed and it does depend on k. Since k is initially unknown, how do you actually parameterize the algorithm? is there a stopping condition that can be tested?
6- In Thm 3.4 it is said that the algorithm returns the subspace V in time that is polynomial in the main parameters of the problem. Yet, I'm not sure where such complexity comes from. In Alg.1 it seems like the subspace is the direct output of the algorithm, so the complexity is r times the cost of computing the gradient, which according to Lem3.2. is poly(n). Is this the way you finally obtain the complexity?
7- One thing I'm doubtful about is the fact that the result in Thm 3.4 seems to be independent from the depth d and width k_i of the different layers. Some conditions may be implicit in the Asm.1 and 2, though. Furthermore, in the experiments it is clearly showed that thin NNs may make the support not recoverable. Could you please make such limit more explicit in the theory?
8- In alg.4 I think lines 5-7 are just the way to execute line 4. Is that correct? If not, how do you execute line 4?
9- In alg.4 line 8 and 9 are not easy to follow and they are not really discussed in the main text. Could you please clarify?
10- The empirical validation is relatively simple but it illustrates quite well the theory. Still I wish the authors could report results that dig more in detail in the theoretical results showing how tight they are (e.g., in the dependency on n, k, and other factors). The current results provide just a hint on how accurate/informative the theory is.
11- In the empirical result, it would be great to have a much more thorough validation of the difference between the non-adaptive and the adaptive algorithms. In the current results it seems like there is very limited difference.

**Experience Assessment:**

I do not know much about this area.

**Review Assessment: Checking Correctness Of Derivations And Theory:**

I assessed the sensibility of the derivations and theory.

**Review Assessment: Checking Correctness Of Experiments:**

I assessed the sensibility of the experiments.

**Review Assessment: Thoroughness In Paper Reading:**

I read the paper at least twice and used my best judgement in assessing the paper.

---

> ### Author Response · Authors · 2019-11-15
> **Response to Reviewer #3**
>
> We thank the reviewer for the comments on our paper. We reply to detailed comments in order below.
>
> 1) We agree that assumptions 3 & 4 are perhaps stronger than the others. For 3, recall that: sigma_V(g) = sigma( A(I-P_v)g), so the worst case is when V contains all but one vector in the span of A, so (I-P_v) is a single direction, and here V will represent the subspace that we have recovered so far. Now if it is the case that tau(sigma( A(I-P_v)g)) is nearly always equal to 0, then we can essentially learn nothing about the behavior of the network in this direction, which motivates the need to have an assumption like this. In terms of practicality, if it were the case that the network always evaluated to 0 in this direction, then the network is in some sense not very sensitive to this direction, and thus it may be acceptable to not recover this direction.
>
> For 4, this condition is arguably much more natural, despite in appearance seeming more complicated. In essence, the assumption just states that if we are given a random direction g such that sigma(cg)> 1, then the directional derivative is not arbitrarily small around the boundary where sigma(cg) passes through 1. Since sigma(0) = 0, the derivative must be non-zero along this direction, and this condition simply asserts that a non-trivial fraction of the velocity in this direction occurs around the boundary (with good probability over the random direction). Notice that if this is not the case, we cannot hope to recover information about this direction, since our only interface to the network comes at the point where we cross the boundary.
>
> 2) We have made several edits and clarifications to address the clarity of the paper.
>
> 3) Thanks for pointing this out. In essence, our gradient computation is done via finite differences and since ReLU networks are piecewise linear, there is no approximation error. Some clarification has been added to the start of Section 3.1
>
> 4) The claim in question is the claim that the “gradients of the network all lie in the row span of A”. We have clarified this section to make it clear what this line is referring to.
>
> 5) If k is unknown, we can always replace k with an upper bound k* on k, and then our algorithm will make r = (k* log(k*)/gamma) queries in alg 1, and we will still recover k/2 vectors (we can only recover more by adding more gradients). Conversely, by the same argument given in the paper, if we have any value t and we make r = (t log(t)/gamma) queries, we will recover at least min{k/2,t/2} vectors in the row-span.
>
> 6) This is correct. Indeed, as in Alg. 1, it is true that the subspace is the direct output of the algorithm, so the complexity is r times the cost of computing a gradient (times log(1/delta) if we run the algorithm log(1/delta) times). Since r=(k log(k)/gamma), and one gradient computation can be done in O(n) queries in poly(n) time, the resulting runtime follows.
>
> 7) Thanks for the question. Our results hold independently of d and k_i as long as the assumptions 1 and 2 hold, which is a stand-alone result that we hope is interesting. However, as in Lemma 3.1, these assumptions only hold when the depth is not too large and the width not too small. Therefore, our experiments support our theory.
>
> 8) This is actually not the case: line 4 computes a value x such that sigma_v(x) is very close to 1 (as in Prop 4.1). This is done by a binary search on values of tau(sigma_v(c*g)) for different values of c. We will clarify further in the algorithm box that this binary search is occurring on line 4. Lines 5-7 perform a second binary search step, this time in other random directions beginning at the point x and moving.
>
> 9) These lines show how one can actually recover the gradient given the values c_i found from in lines 5-7. Since the proof and a thorough discussion of this is somewhat technical, due to space constraints it was difficult to fit an informative explanation, however we have added some clarifications on these steps in the paragraph before Lemma 4.2. Roughly, the point is that because of the thresholding tau, the only values we know are the scalings c_i, which tell us how far one must go beginning at x in the direction of u_i before the boundary sigma(x + c_i u_i) = 1 is reached. So 1/c_i is proportional to $\nabla_{u_i} \sigma(x)$.  We compute n of these values c_i in linearly independent directions u_i. Since the directional derivative is the dot product of u_i and the gradient y^*, we can set up a linear system <y,u_i> = 1/c_i for i=1,2,...,n and solving this linear system will give us the gradient y^*.
>
> 10) Thanks for the suggestion. We do not attempt to produce tight but simply reasonable bounds in our theorems. Therefore, no tightness experiments were done. However, we believe our experiments demonstrate that our bounds are on the right order of magnitude.
>
> 11)Thanks for the suggestion. We seem to have observed a limited difference as well and believe that this is the case.

---

### Official Review · AnonReviewer2 · 2019-10-28
**Official Blind Review #2**

**Rating:** 8

**Review:**

Summary: The paper considers the problem of recovering the span of the latent variables of a neural network with various activation functions. More precisely, if we write a neural network as M(x) = f(Ax), for some neural network f, and a matrix A: R^{k x d} -> R^{k}, k << d, we wish to recover the row span of A. The authors consider ReLU activations -- in which case they can recover at least "half" of the row span with ~kd queries to M(x), and smooth activations -- in which case they can approximately recover the row span in poly(k,d) queries. The authors also consider (empirically) applications of these algorithms to "input obfuscation": namely generating samples which are effectively noise, but the network classifies them as "structure" (e.g. digits on MNIST).

Evaluation: This is a strong submission. The paper is well written, easy to follow, and contains various interesting techniques, possibly for a wide audience in ICLR. For instance, the ReLU algorithm relies on the piecewise affine structure of ReLU nets to reduce calculating gradients to solving simple linear systems; it additionally cleanly characterizes how many "sign patterns" need to be seen to span most directions of the gradients of M. The differentiable activation case also has lots of neat tricks for dealing with non-linearity, and in particular how to find the right "scaling" of directions to move in to get new almost orthogonal information about the current estimate of the row span.

**Experience Assessment:**

I have published in this field for several years.

**Review Assessment: Checking Correctness Of Derivations And Theory:**

I assessed the sensibility of the derivations and theory.

**Review Assessment: Checking Correctness Of Experiments:**

I assessed the sensibility of the experiments.

**Review Assessment: Thoroughness In Paper Reading:**

I read the paper at least twice and used my best judgement in assessing the paper.

---

> ### Author Response · Authors · 2019-11-15
> **Response to Reviewer #2**
>
> We thank the reviewer for the comments on our paper.

---

### Official Review · AnonReviewer1 · 2019-11-03
**Official Blind Review #1**

**Rating:** 6

**Review:**

This paper is interesting to me in terms that it provides a systematic approach to generate adversarial samples for a given black-box neural network system. Though this is the side product of the paper.

Some questions:
1.	Finding adversarial examples in this paper relies on finding the null space of Ax =0. This requires that input data is with a higher dimension than the span of A. I understand that the whole paper assumes that n>k where input is with dimension n and A is k by n. However, this restricts the application of the proposed adversarial attack as a general approach.
2.	When we are trying to recover the span of A, how can we judge if or not M(.) has differential activation functions? Which algorithm (1 or 2) should we try?
3.	Does the theorem rely on the assumption that A is with rank k? In general, A^{k by n} does not guarantee to be with rank min(k,n). For example, people may use low-rank matrix factorization to approximate the weight of some layers during neural network training.


**Experience Assessment:**

I do not know much about this area.

**Review Assessment: Checking Correctness Of Derivations And Theory:**

I assessed the sensibility of the derivations and theory.

**Review Assessment: Checking Correctness Of Experiments:**

I carefully checked the experiments.

**Review Assessment: Thoroughness In Paper Reading:**

I read the paper at least twice and used my best judgement in assessing the paper.

---

> ### Author Response · Authors · 2019-11-15
> **Response to Reviewer #1**
>
> We thank the reviewer for the comments on our paper. We reply to the three questions in order below.
>
> 1) For the first question, we believe that many practical networks that might be attacked satisfy k < n since the dimension of the input is often large and increasing the input dimension could be inefficient.
>
> 2) Thanks for this interesting question. The difference between Algorithms 1 and 2 might not have been clearly specified: this has now been edited and clarified. In general, Algorithm 1 is used to attack networks with no thresholded output (i.e. regression) and Algorithm 2 is for networks with a thresholded output (i.e. classification). Note that without a threshold function, if the network is differentiable then the problem of recovery becomes strictly easier, and we can apply algorithm 2 anyway, but we can replace the steps in Algorithm 2 which compute the gradient by simply computing the gradient by finite differences. So if the output of the network is thresholded, then Algorithm 2 should certainly be applied.
>
> Otherwise, in practice we can just run both algorithms and use the one which recovers the larger subspace (i.e., take the one which recovers more vectors in the span). Since precision loss will be incurred for finite differences if the activation functions are not piece-wise linear, one can just throw out vectors v which are very close to lying in the span of previously recovered vectors (since such a vector v may actually be in this span, but is not due to precision loss).
>
> 3) Our theorems do not depend on the fact that A is rank k and extend to the case when the matrix is not full rank. A clarifying sentence has been added.

---

### Official Review · AnonReviewer4 · 2019-11-03
**Official Blind Review #4**

**Rating:** 3

**Review:**

The paper studies the problem of span recovery for deep neural networks, that is
recovering the space of inputs that affect the output of the network.
The authors propose two algorithm, one for the case of ReLU activations and one
for the case of differentiable activations, and theoretically prove that they can
recover the span under certain assumptions. They complement these results with
experiments on a simple model on MNIST.

At a high level, the algorithms rely on computing gradient information through
finite differences in order to recover direction in input space that the model
is sensitive to. The assumptions are reasonable, mostly requiring that when
passing random inputs through the model, the output varies sufficiently while
the gradients are sufficiently large with some probability. Given these
assumptions, the algorithms maintain a subset of the span and iteratively find
directions in the input space that are not captured by the current subspace.

Overall, the paper is well-written and addresses a fairly fundamental problem.
The solution presented is motivated and the analysis intuitive. However, I am
not fully convinced about the importance of studying span recovery in this
setting. In a white-box setting, we can just run SVD on the first layer. The
application to input obfuscation is fairly unconvincing since we already know
how to perform query-only adversarial attacks. Finally, I have a few questions
about the experimental results (see below).

I thus recommend weak rejection at the moment but would be willing to
reconsider based on the author response.

Specific comments:
-- Algorithm 2 relies on the value of sigma when only tau(sigma) is available.
This should be clarified and the specific lemmas that allow this referenced.
-- The first sentence of the experimental section is confusing. When is this
computation performed? Based on the plot, the matrix is always full rank (80).
-- More generally, how is the evaluation performed? Is the span returned
compared to the ground truth (via SVD)?
-- I find it somewhat odd that it is possible to recover a rank 80 subspace with
100 samples. After all, the subspace is described by 784 * 80 reals (which is
also the theoretical complexity O(n * k)) and we are only allowed input-output
queries.  Can the authors provide some intuition/clarification? Are these a 100
_gradient_ computations?
-- Can the authors provide additional details about how they compute gradients
experimentally (using finite differences)?
-- Why are the input obfuscation experiments not performed with the subspace
recovered by the proposed algorithms? This would be necessary to argue that
partial recovery actually leads to adversarial vulnerability.
 -- Denoting a neural network by sigma in the abstract and first intro paragraph
 is confusing (since sigma denotes the activations later).
-- Prop 4.1: gradient needs a norm.


##### Post-discussion update #####

Thank you for your response.
I do agree that input obfuscation is somewhat different from adversarial attacks. It would be interesting to see if it has any applications to more realistic scenarios.

To be honest, I am somewhat disappointed to find out during the discussion that full gradient computation (i.e., not the proposed algorithm) was used for the empirical results. Most of the paper is focused on proving that finite differences are sufficient, yet the empirical results completely bypass this difficulty. I will thus keep my score the same.

**Experience Assessment:**

I have published in this field for several years.

**Review Assessment: Checking Correctness Of Derivations And Theory:**

I assessed the sensibility of the derivations and theory.

**Review Assessment: Checking Correctness Of Experiments:**

I carefully checked the experiments.

**Review Assessment: Thoroughness In Paper Reading:**

I read the paper at least twice and used my best judgement in assessing the paper.

---

> ### Author Response · Authors · 2019-11-15
> **Response to Reviewer #4**
>
> We thank the reviewer for the comments on our paper. For the first comment about the white vs black-box setting, we agree that span recovery is useful mainly in the black-box setting. In this paper, we work in that setting and only assume query-based information. We note that span recovery can be used to aid a black-box attack, as demonstrated by the input obfuscation attack on MNIST networks. When compared to existing query-only adversarial attacks, we believe that input obfuscation can be a useful context to consider since adversarial attacks aim to drastically change the output while keeping the input similar; however, input obfuscation attacks aim to drastically change the input while keeping the output similar.
>
> We reply to the specific comments in order.
> 1) Indeed, we are only allowed to query for values of tau(sigma) and not sigma, and so this line (line 4) should be clarified further. As mentioned in the comment next to line 4, the value x that is found here is a result of proposition 4.1, wherein we discuss the details of the procedure which actually obtains x. To find x, we choose a random direction g (as shown in the algorithm box), and then we just do a binary search over the values of tau(sigma(c*g)) for different values of c until we find an x = c*g such that sigma(x) is as close as possible to 1. We added some details in Algorithm 2 to make it clear that this binary search is taking place.
>
> 2) Thank you for pointing out that the first sentence of the experiments was confusing, this sentence has been edited to be more clear. The computation of the gradient matrix is performed after a fixed number of gradients have been evaluated and then the rank of the matrix is calculated. The matrix is always full rank after a certain number of gradient samples (around 100) since as our theory predicts, most gradients are linearly independent.
>
> 3) The evaluation is performed by comparing the rank of  the recovered gradient matrix to the rank of the true first-layer weight matrix, of which we are recovering the span. This is described as full span recovery, where the rank of the recovered span is equal to the rank of the underlying matrix. The span, itself, is not directly compared since the gradients lie in the span of the first-layer weight matrix. However, our MNIST attack experiments demonstrate that the span recovered is indeed correct.
>
> 4) Thanks for pointing out the confusion. We use gradient computations and each sample is indeed a full gradient computation. This has now been edited and clarified.
>
> 5) We use gradient computations via auto-differentiation but since our networks are ReLU activated and piece-wise linear, there should be no loss of precision when using finite differences. This has now been edited and clarified.
>
> 6) We do in fact perform our input obfuscation with the subspace recovered by our algorithm, which demonstrates adversarial vulnerability of the network. In the case of our experiments on the MNIST network, our attacks fully recover the ground-truth span of the network (the recovery is without loss of precision since the network is ReLU activated and piece-wise linear). So in this case, the subspace recovered by our algorithm and the underlying subspace are the same. Thus, our experiments suggest that in practice, it is possible to obtain full span recovery, as opposed to the partial recovery which we can prove theoretically in Theorem 3.4.
>
> 7 + 8) This is a good point about the sigma notation, we changed the notation here to be consistent with the rest of the paper. Specifically, sigma should usually refer to a full network in this paper, so we changed the other activation functions to phi. We also added a norm to the gradient in Proposition 4.1.

---

### Decision · Program_Chairs · 2019-12-19

**Decision:**

Accept (Poster)

**Comment:**

The authors propose a way to recover latent factors implicitly constructed by a neural net with black box access to the nets output. This can be useful for identifying possible adversarial attacks. The majority of reviewers agrees that this is a solid technical and experimental contribution.